# Smart Crop Cultivation System Using Automated Agriculture Monitoring Environment in the Context of Bangladesh Agriculture

**DOI:** 10.3390/s23208472

**Published:** 2023-10-15

**Authors:** Md. Bayazid Rahman, Joy Dhon Chakma, Abdul Momin, Shahidul Islam, Md Ashraf Uddin, Md Aminul Islam, Sunil Aryal

**Affiliations:** 1Department of Computer Science and Engineering, Faculty of Science and Technology, Notre Dame University Bangladesh, Dhaka 1000, Bangladesh; md.bayazid.rahman.25.07@gmail.com (M.B.R.); joydhon.ndub@gmail.com (J.D.C.); 2Agricultural Engineering Technology, School of Agriculture, Tennessee Tech University, Cookeville, TN 38505, USA; momin@tntech.edu; 3Department of Agriculture, University of Arkansas at Pine Bluff, Pine Bluff, AR 71601, USA; 4School of Information Technology, Deakin University, Geelong, VIC 3220, Australia; ashraf.uddin@deakin.edu.au (M.A.U.); sunil.aryal@deakin.edu.au (S.A.); 5Department of Computer Science and Engineering, Jagannath University, Dhaka 1100, Bangladesh; aminul@cse.jnu.ac.bd

**Keywords:** crop yield prediction, smart agriculture, decision tree classifier, crop monitoring, irrigation control

## Abstract

The Internet of Things (IoT) is a transformative technology that is reshaping industries and daily life, leading us towards a connected future that is full of possibilities and innovations. In this paper, we present a robust framework for the application of Internet of Things (IoT) technology in the agricultural sector in Bangladesh. The framework encompasses the integration of IoT, data mining techniques, and cloud monitoring systems to enhance productivity, improve water management, and provide real-time crop forecasting. We conducted rigorous experimentation on the framework. We achieve an accuracy of 87.38% for the proposed model in predicting data harvest. Our findings highlight the effectiveness and transparency of the framework, underscoring the significant potential of the IoT in transforming agriculture and empowering farmers with data-driven decision-making capabilities. The proposed framework might be very impactful in real-life agriculture, especially for monsoon agriculture-based countries like Bangladesh.

## 1. Introduction

Bangladesh is called an agriculture-based country, and its economy mainly depends on crops such as grains, vegetables, pulses, oilseeds, and tubers. About 50% of the population of Bangladesh primarily depends on this agriculture-based sector, and more than 70% of the land is suitable for agricultural use [1]. Farmers often grow rice, jute, wheat, tea, legumes, oilseeds, vegetables, and fruits in this country. The soil in Bangladesh is ideal for growing a wide variety of crops. Although agriculture significantly contributes to Bangladesh’s economy, crop yields remain uncertain, and farmers have yet to fully commercialize their cultivated products due to their reliance on traditional methods. Bangladesh has a total land area of 14,570 km^2^, and 60% of this land is arable [2]. However, with the population increasing at a rate of 1.37 percent per year, the amount of arable land is decreasing daily. Unchecked urbanization, industrialization, and human activity are causing agricultural areas to disappear gradually. Nations with economies reliant on agriculture should embrace modern technologies to enhance crop productivity and implement sustainable farming practices. Strengthening agricultural production systems is vital to increasing income and ensuring food and nutrition security in underdeveloped nations [3]. Bangladesh’s government has recently undertaken an initiative to embrace the IoT’s fourth industrial revolution in the agriculture sector under its a2i program. A recent study [4] suggests that the adoption of the fourth industrial IoT might potentially improve irrigation efficiency by up to 50% by the year 2030 [4].

IoT technologies are widely used in the agriculture sector in many developed countries to increase production and meet the demand for food supply in the market. IoT in agriculture can reduce production costs and time by providing precision agriculture.

Farmers very often encounter financial losses resulting from unforeseen natural calamities. However, with access to advanced weather forecasts through IoT technology, they might be able to avoid or mitigate these losses to a certain extent. By incorporating IoT solutions, farmers can receive real-time weather forecasts and remotely monitor their agricultural operations, which enables them to make informed decisions accordingly. Similarly, our proposed framework can empower farmers to visualize sensor data, control irrigation pumps, and optimize plant and water management practices for improved productivity and resource efficiency. Many state-of-the-art projects combining IoT and data mining techniques in the agriculture sector have been carried out to develop smart agriculture infrastructure [5]. The application of IoT in agriculture has brought great revolutionary changes to the agricultural environment by addressing multiple challenges and examining different complexities [6]. Our research targets the adoption of an IoT monitoring system for farmers to solve problems such as water crises, cost management, and productivity issues [6,7].

However, advanced technology is usually, but not always, beneficial to humans. There needs to be careful research on how to develop green technology for the survival of humanity and the animal kingdom. We need to avoid developing destructive technology that endangers people and the earth itself. Considering this issue, we developed an environmentally friendly monitoring system that helps the farmer provide information about the source and characteristics of the grain or product. In this system, we obtained data for predicting crops’ transplanting and harvesting times from wireless sensors consisting of IoT devices. This system facilitates monitoring and controlling the water supply on the land automatically.

Our main contributions to the paper are as follows:To the best of our knowledge, our work represents one of the first frameworks including IoT and cloud monitoring systems in a monsoon climate, such as that found in Bangladesh. Through our system framework, multiple forecasts can be made available to the farmers by evaluating the cloud-based data;We propose two algorithms, where one, describes the procedure of collecting data from sensors through our microcontroller and sending it to the cloud system, and the other is designed to forecast crop planting and harvesting times with duration. Our database system is developed as a web-based application to enable easy interaction for end users, particularly farmers, with our proposed system. The stored data serves as a valuable resource for automated decision-making in crop cultivation, particularly in water control measures.

By integrating web applications, our system can facilitate the evaluation of data from IoT devices in Bangladesh, providing farmers with clear insights into IoT device performance. While Bangladesh’s environment and agricultural products might be unique, neighboring Southeast Asian countries such as Bhutan, Nepal, India, and Myanmar can adopt our suggested approach using their statistics. Moreover, with minor adjustments, our flexible framework can be implemented in agriculture in any country.

Subsequent sections of this paper are arranged as follows: Section 2 explains the background. Section 3 discusses methodology, system design, and overview and presents how the system has been implemented. This section also has two algorithms: (I) crop data generation and (II) harvesting and transplanting time forecasting. Section 4 reports all research and analysis of result data by data mining focused on crop cultivation. Section 5 presents our discussion and future work on our research, and finally, we summarize the conclusions in Section 5.

## 2. Related Work

Fan et al. [8] proposed a system framework for establishing an intelligent agriculture platform using big data analysis and IoT sensor data via cloud technology. Andreas et al. [9] provide a thorough review of big data analysis in agriculture, analyzing thirty-four research papers to identify the current applications, challenges, and potential solutions. Their work highlighted the increasing availability of big data sources, tools, and techniques that can drive innovation and research for smarter farming practices, ultimately contributing to sustainable agriculture and higher-quality food production. A system structure was developed in the article [10] to improve the combination of big data and artificial intelligence in agriculture, where data from IoT sensors was received and stored in the cloud to monitor the farm. They created a control system based on data management and node sensors in crop fields for smartphones and online applications. In the article [11], a system framework was created and built. The framework consisted of three components: a control box, a web application, and a mobile application. Their method was put in place to regulate crop irrigation and govern agricultural plots. The solenoid valve switching procedure by the farmer is controlled by a smartphone app. A survey of the literature was centered on studies and analyses of the application of IoT in modern farming [12]. Their research and analysis showed how China can reduce human effort in agriculture by relying on IoT technology. They presented some categories by analyzing agricultural system development. By explaining the architecture and applications of cloud technologies, the researchers in [13] focused on the importance of using IoT and cloud computing in the agricultural sector. This layered architecture, in conjunction with Radio Frequency Identification (RFID) technology, is used to automate planting and production. Doshi et al. [14] proposed an IoT technology that generates messages from their applications to instruct farmers to suggest smart farming.

As surveyed in the scientific article [15], IoT has been used in a variety of investigations in recent years. They reviewed modern farm technology and explored a variety of live monitoring systems for IoT-based applications and wireless sensor networks. They also discussed well-known technologies that are continually pushing the IoT to improve. They also listed some of the obstacles we may face when working in agriculture with IoT, including hardware constraints, networking challenges, technical concerns, resource optimization, and mobility. This systematic review [16] delves into the integration of cutting-edge technologies like predictive modeling algorithms, deep-learning-based sensing, and big urban data in shaping immersive digital twin cities. By analyzing the recent literature, the paper establishes the significance of virtual simulation tools, spatial cognition algorithms, and multi-sensor fusion technology in developing sustainable urban governance networks and data-driven smart city environments. The study provides valuable insights into the role of the Internet of Things, digital twin modeling, and intelligent sensing devices in building smarter and more connected urban infrastructures. The work proposed by Nandan et al. [17] provides a literature review by illustrating how climate change affects the agriculture and food security of the Barisal district in Bangladesh. Here, they discussed the environmental condition of the Barisal district and the impact of rainfall, drought, waterlogging, thunderstorms, excessive fog, and climate change on agriculture production. The authors of [18] presented constructive research on the overall status of technology-dependent agriculture in Bangladesh. A quality-aware autonomous information system for agriculture services based on agriculture-related data was developed in the article [19]. A literature review on the role of Internet of Things technologies in agriculture that explored the varied effects of IoT in agriculture, the benefits and drawbacks of IoT devices, and the application layer required for farming in current technology was introduced in article [20]. The authors of [21] suggested a smart agriculture system design that enhanced a smart farming system for effective management and control of agricultural greenhouses through IoT and data mining technology to increase production in agriculture. They employed IoT technology to collect a large amount of environmental information from grain greenhouses and used advanced algorithms to pick relatively favorable data as a clustering method for environmental reference data.

Thomas et al. [22] addressed the various systems, frameworks, and multiple sources for smart farming. They emphasized the utilization of cloud computing and big data technology in the development of existing agricultural event systems. An alert system was proposed in [23] that presents a system framework capable of controlling the amount of water passing through IoT devices in agriculture. Said et al. [24] proposed a method to determine the minimum amount of irrigation and the maximum amount of water used on the plants through an intelligent irrigation plan. By keeping an eye on the water position and irrigation schedule of the tomato crop in extremely dry climate conditions, this approach sought to investigate the efficacy of the Intelligent Irrigation System (IIS) related to Water Use Efficiency (WE) and Irrigation Water Use Efficiency (IWU) and determine its viability.

The authors of [25] discussed a proposed framework that aims to balance energy efficiency and security in precision agriculture. The framework uses hashing as the only form of advanced encryption, which adds an extra layer of security to the public channel. Unlike existing management systems, this proposed method does not store public keys. By allowing on-field sensors to not be directly connected to the sink node, the proposed system provides significant residual energy savings. Compared to the current aggregation strategy, the suggested scheme results in about 35% more alive nodes and 32% greater retention of residual energy. The authors of [26] proposed a trust management approach for ensuring the security of smart agriculture in the cloud-based Internet of Agriculture Things (IoAT). The authors suggest that the integration of cloud computing with the IoAT can significantly improve the efficiency of agriculture, but it also poses security challenges such as data privacy, integrity, and authenticity. The AgriTrust approach (a trust management mechanism that substitutes for conventional cryptography methods) consists of three main components: a trust model, a trust evaluation mechanism, and a trust management mechanism. The trust model defines the trustworthiness of entities in the IoAT, such as devices, sensors, and cloud servers. The trust evaluation mechanism is used to evaluate the trustworthiness of entities based on their past behavior and feedback from other entities. The authors of [27] propose an IoT-based WSN framework that provides an efficient and secure solution for smart agriculture applications. The proposed scheme’s use of a hierarchical architecture, data aggregation and compression techniques, and secure data transfer protocols can significantly improve the efficiency and security of smart agriculture applications. The proposed framework consists of three tiers: the sensor layer, the intermediate layer, and the application layer. Additionally, they have proposed a secure data transfer protocol that makes use of Elliptic Curve Cryptography (ECC) and the Advanced Encryption Standard (AES) to ensure the security of the data transmitted between the sensor and the application layer.

## 3. The Proposed Framework

We have implemented our real-time cloud-based web and mobile applications integrated with IoT sensors. We have also analyzed our previously collected agricultural data using different data mining techniques.

Next, we have described IoT sensors for our real-time application, followed by a discussion about our analytical approach for previously collected raw data. Finally, we have described the operations of our application for providing real-time decisions based on analyzed data. Figure 1 depicts a summary of our suggested approach as a whole. The crop field sensors gather data, pre-process it, and send it to the cloud database. Farmers are given access to real-time cloud-based data and users via mobile and online applications via the application from the cloud database, which is depicted step-by-step in the following figures.

In many precision agriculture studies, algorithms are trained and tested using publicly available datasets, which might not accurately represent real-world scenarios due to varying soil and weather conditions across countries. To address this concern, we develop our precision agriculture algorithms using a dataset collected from IoT devices deployed in actual crop fields. This ensures a more realistic and relevant evaluation of our approach. We present the detailed steps involved in our algorithms designed for precision in Algorithms 1 and 2.

### 3.1. Design and Overview of the System

In this section, we demonstrate the functionalities of our system, which consists of IoT devices, cloud databases, and websites or mobile apps. In this framework, we send data from the farm to the web application or smart phone through the data management controller named NodeMCU. An open-source electronic platform called NodeMCU is built around the ESP8266 Wi-Fi system-on-chip (SoC) [28,29]. This enables the simple development of Internet of Things (IoT) projects by coupling a microcontroller unit (MCU) with integrated Wi-Fi capabilities. The design and implementation overview of this system is divided into three components: hardware, web/mobile applications, and cloud databases, as shown in Figure 2.

With reference to Figure 2 and Figure 3, the initial component was created as a control box. This control box transmits data from sensors and manages IoT devices. The control box includes an automated water pump regulator, a DHT 11 (Temperature and Humidity Sensor Device Model) sensor, a NodeMCU, and a soil moisture sensor. A program input is provided to the NodeMCU through which we collect data from IoT sensors and send it from the control box to the cloud database connected to the web and mobile applications. The concluding two figures illustrate a real-time and test area perspective of this control box configuration.

Then the second component is the web application http://smart-farming.com.bd/ (accessed on 8 January 2023) that is represented by Figure 2 and Figure 4. Mobile applications also have similar configurations as web applications. The mobile application was developed by MIT App Inventor Web as a dummy mobile application that has not yet been published anywhere. MIT App Inventor (https://appinventor.mit.edu/, accessed on 14 August 2023) is a web-based visual development environment that allows users to create mobile applications for Android devices without requiring extensive programming knowledge. It provides a graphical interface where users can drag and drop components, define their behavior using blocks, and build fully functional apps. The cloud database, which is the third and final component, is crucial in maintaining the data gathered and organizing the data from the database on the application. The node of the first component in Figure 2 sends the data from the NodeMCU to the cloud through the API (Application Programming Interface) and stores it in our third component’s cloud database. Farmers can check the old data and analyze their own farmland to see what kind of crop can be produced.

This whole system consists of seven parts, as shown in Figure 5, which are: (i) physical layer: in the physical layer, our physical devices like wireless sensors, solenoid bulbs, relay modules, analog signal modules, and power sources exist; (ii) link layer: in the link layer, we have used the Wi-Fi (Wireless Fidelity) network as the coordinator sensor node, which connects the devices of the encapsulation layer with the devices of the physical layer; (iii) encapsulation layer: in this layer, we use a NodeMCU device that is able to communicate with IPv6 (Internet Protocol version 6) in the network security system; (iv) middleware layer: through this layer, we acquire the data taken from the agriculture land and environment and send it to the configuration layer; (v) configuration layer: the configuration layer basically collects and analyzes the raw acquired data and sends the data to its destination in the configuration layer; (vi) management layer: This layer combines previously acquired data with newly analyzed data, which gives results. On the basis of that result, the prediction report and other information are managed; and (vii) application layer: by presenting the data in an organized manner to the end user or farmers through the application layer, the farmers today can decide what to do next.

### 3.2. System Implementation

The proposed approach transmits temperature and humidity information using DHT11 and soil moisture sensor devices to the cloud database via a web or mobile application using a microcontroller named NodeMCU integrated with an ESP 8266 module. We utilized a NodeMCU that uses a lot less electricity. The NodeMCU web server also uses 60% less power with a latency of 1 ms [30]. Additionally, the remaining sensors and equipment employed in the physical layer are extremely energy-efficient. They do not consume a lot of electricity. The solenoid valve or irrigation pump is automatically turned on and off depending on the soil moisture sensor data. Additionally, this on-off status is immediately sent to our application by NodeMCU. If the amount of water in the soil decreases, then the solenoid valve opens automatically, and if the amount of water in the soil becomes balanced according to the program set in the microcontroller NodeMCU, then the solenoid valve automatically shuts off. The application automatically analyzes the data from the cloud database and provides monthly crop harvesting and planting information. In the context of Bangladesh, according to our study, almost all the crops have a planting time of about 3 months [31]. This mobile or web application has crop calendars [32] so that the farmers can know about their crop planting, lifting, fertilizer application, and water quantity properly. At present, about 33 types of crop data have been collected in our cloud database. A few of the top scientific publications included in Table 1 are contrasted with some of the features included in our proposed system for farmers.

#### 3.2.1. Crop Data Generation

We collect sensor data from IoT sensor devices through the ESP8266 NodeMCU and send it to the cloud, as shown in Algorithm 1. We set up an internet connection through wireless technology using the ***wifi_setup()*** module. To do this, it is essential to know the *Service Set Identifier (SSID)* and *Password* of the wireless router to connect the cloud application with the NodeMCU microcontroller. The ***read_soil_moisture_sensor()*** function and the ***read_temp_hum_sensor()*** function collect soil moisture, temperature, and humidity data and send them to the cloud. The ***irrigation_control()*** functions mainly work to ensure the correct amount of water. This basically ensures the amount of water is precisely controlled through irrigation bulbs or water pumps if the amount of moisture in the soil is low. This function is conditioned in such a way that if the ***sms_data*** obtains less moisture in the soil, it will automatically release water from the water pump, and if the moisture level is at the correct level, then the water pump will automatically stop.

The algorithm aims to maintain optimal soil moisture for enhanced crop yield. This is intelligently set based on soil moisture levels and weather conditions. Our algorithm aids in conserving water resources by intelligently adjusting irrigation in the crop field. The algorithm combines advanced sensing technology with intelligent water control techniques, making it a promising solution for precision agriculture. Our algorithm can potentially address water scarcity challenges and enhance agricultural productivity.
**Algorithm 1:** Obtain data from DHT 11, the soil moisture sensor, and control irrigation
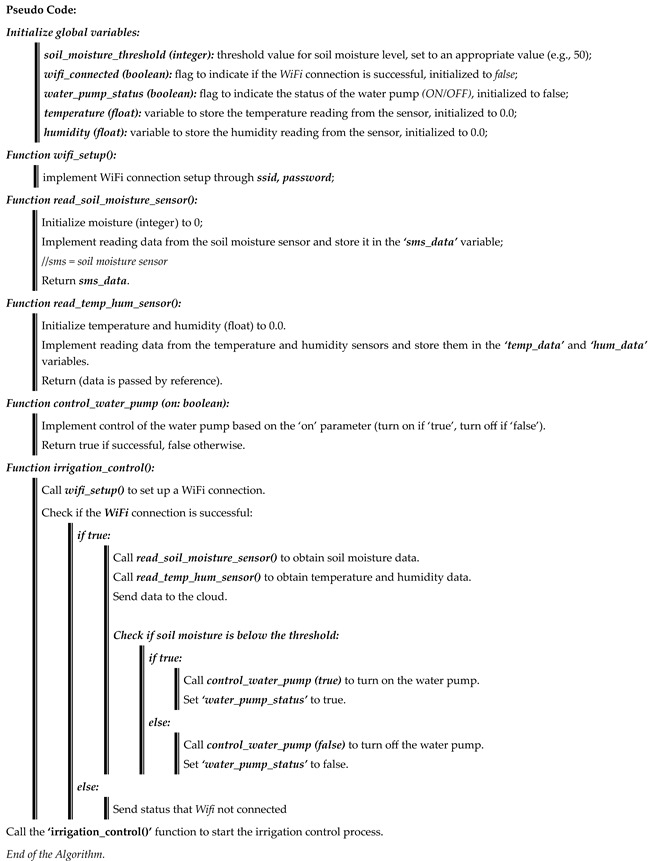


#### 3.2.2. Harvesting and Transplanting Time Forecasting

In our Algorithm 2, we take the data from the physical device sensors, present the real-time data to the farmers, compare the real-time data with the already collected training data (examined by data mining), and present the analysis to the farmers by dividing the crop planting and harvesting data into two parts. The ***server_setup()*** function is implemented for cloud server connections. ***Gate_field_data()*** and ***cross_data()*** functions collect data from physical devices and show real-time data to farmers. The ***crop_analysis()*** module shows farmers a real-time forecast of what crops can be planted and harvested in the current month and by when. By comparing our already collected crop data with the data we are obtaining in real time, the output is shown to the farmers.
**Algorithm 2:** Forecasting of crop harvesting, transplanting crops with duration, and crop analysis
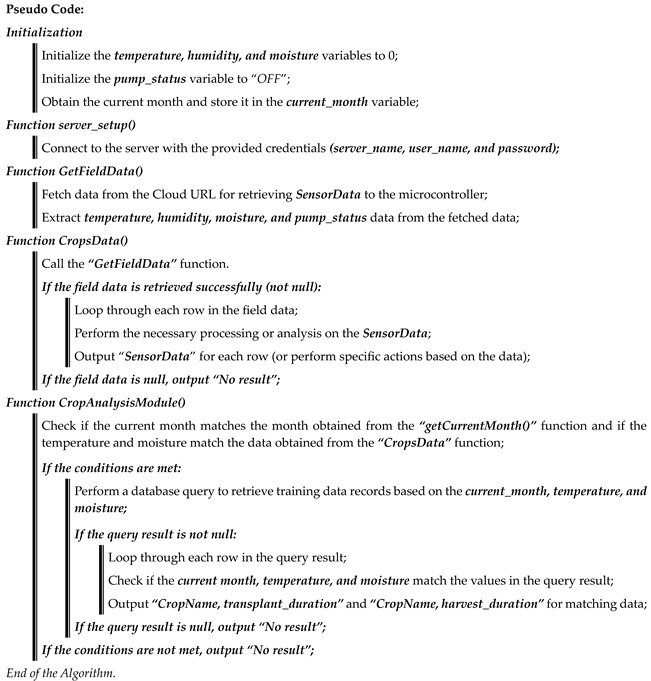


With the Crop_Analysis_Module, the farmer himself will be able to know the time of transplanting and harvesting by analyzing the algorithm with the month, temperature, and humidity as input.

Unlike traditional methods, which rely on simple data processing, this algorithm leverages historical crop data and machine-learning models to enable data-driven decision-making and achieve more accurate and efficient crop analysis. After pre-processing data from sensors with trained models, the algorithm offers insights into crop health, growth anomalies, and optimal crop management strategies. This innovative approach represents a significant advancement in the field of precision agriculture, enabling farmers to make informed decisions that maximize yields and resource utilization while reducing environmental impact.

#### 3.2.3. Data Analysis

This subsection demonstrates how we have performed the work of data analysis. Figure 6 shows the processing steps of data analysis.

Pre-processing

The quality of the data determines the quality of the knowledge; hence, this is a crucial phase in the knowledge discovery process. Since the data for temperature and humidity are numerically large and the data for each row is different, we normalize them by the following formula [33]:(1)$$X_{new}=\frac{\left(X-X_{min}\right)\;}{\left(X_{max}-X_{min}\right)},$$
*where X = the collection of observed values found in X, X_min_ = the minimum value in X, and X_max_ = the highest value in X.*

II.Clustering

The process of grouping abstract items into several categories is called clustering. When N items are divided into a number of clusters using *K-Means* clustering, each object is a part of the cluster that is closest to the cluster’s centroid. This method highlights the maximum possible differences between the different *k* clusters.

After normalization, we go to the second step and run the dataset using the *K-Means* cluster algorithm. In our experiments, we have used *k* values from 1 to 9 empirically to find out the closet centroid using elbow methods. The following step with the selected cluster *k* will be taken after extracting the value of each centroid using formula (2) to determine the value of the most ideal *k* value using the elbow method and our class attribute. The Sum of Squared Error (SOSE) in Equation (2) and Incorrectly Clustered Instances (ICI) in Equation (3) of each cluster were covered in Section 4.4 (Data curation and result).

The Sum of Squared Error (SOSE) is expressed as:(2)$$SOSE=\sum_{j}\sum_{i}{\left(Y_{ij}-\tilde{Y_{j}}\right)}^{2};$$
*where Y represents the observed values, *
$\tilde{Y}$
* represents the predicted values, and this is the mean of the values of Y.*

To identify incorrectly clustered instances using *K-Means*, it can be expressed as:(3)$$d_{i}<d_{j},\;\mathrm{for}\;any\;j\;\neq i;$$
*where d_j_ is the distance between a data point and its assigned centroid and is greater than or equal to *
$d_{i}$
*, or if the distance between a data point and the centroid of another cluster is greater than or equal to*
$\;d_{j}$
*, then the data point is deemed to be erroneously clustered.*

The *K-Means* cluster formula is given in Equation (4) [34,35,36]:(4)$$J=\sum_{j=1}^{k}\sum_{i=1}^{n}\|x_{i}^{j}-c_{j}{\|}^{2};$$
*where J = the objective K-Means cluster function, n = the number of instances, k = the number of clusters, and *
$\|x_{i}^{j}-c_{j}{\|}^{2}$
*= the Euclidian distance function.*

## 4. Experimental Setup and Result Discussion

### 4.1. Datasets

Based on crop transplantation data from the yearbook of *Agriculture Statistics Bangladesh 2020* [31], we have collected a total of 241 instances of harvesting time data, temperature data, and crop transplanting time data and fed them into training to build our model. After then, we gathered information about the crop season based on information about the time of crop transplanting from the Bangladesh Crop Season Article [37]. “Season” attribute is maintained as a class feature or attribute in our dataset, which includes a total of 10 features. The data is then pre-processed and assembled into a comprehensive data collection. This dataset includes data on 48 crops from the Kharif-II season, 22 crops from the Rabi season, and 33 crops from the Kharif-I season. Kharif (mid-November to mid-March) and Robi (mid-November to mid-March) are the two distinct monsoon agricultural seasons in Bangladesh. Kharif-I (mid-March to mid-July) and Kharif-II (mid-August to mid-October) are further divisions of the Kharif season (mid-July to mid-November). Winter vegetables, wheat, potatoes, legumes, oilseeds, and boro rice are among the rabi crops. Summer vegetables, jute, Aus and Aman rice, and other crops are typical of the Kharif season. This collection contains information on a total of 103 distinct Bangladeshi crops. Our datasets are available through the cloud-based Mendeley Data repository [38].

### 4.2. Real-Time Data Acquisition Module

In our implementation, both web applications and mobile applications work in the same way, which will make the user feel comfortable. To make it more user-friendly, we have multilingual support, namely Bengali and English language interfaces, and a voice-over command option. Our work is available on the cloud as a website.

### 4.3. Real-Time Forecasting

These elements allow the user to easily update and analyze farmers’ land information. With these components, they will be able to check, update, and analyze real-time data and previous records. Through this web and mobile application, they will obtain accurate instructions on which crops can be planted and harvested in which month. Via our technology, farmers will also be able to examine real-time information about their fields, including the temperature, humidity, soil moisture, and operation of the automatic irrigation pump. The pictorial view of real-time crop and weather forecasts, as well as a few other features in the web-mobile application, are shown in Figure 7 and Figure 8, enabling farmers to regularly monitor all the necessary information of their field from the ease of their homes or anyplace else.

### 4.4. Data Curation and Results

We ascertain the Sum of Squared Error (SOSE) and Incorrectly Clustered Instances (ICI) of each cluster. Table 2 shows the result of the Sum of Squared Error (SOSE) and incorrectly clustered instances (ICI) for each cluster.

A graphical representation of SOSE and ISI for each cluster is shown in Figure 9a,b. From Figure 9a, it can be seen that the SOSE of each cluster from cluster 3 to 6 is between 300 and 390. On the other hand, in the ICI of Figure 9b for each cluster, a sharp increase is visualized from cluster 3 to 4, followed by a steady increment. Additionally, since we are working on the dataset based on the season, we also have three common attributes for the season, which are Kharif-I, Robi, and Kharif-II. Therefore, we decide to keep *k = 3* as our clustering parameter for our dataset.

### 4.5. Data Modeling

From the prepared information, knowledge is derived at this stage. Data modeling and discovery typically employ clever techniques to find patterns in the data. Classification, grouping, relationships, and other analysis techniques are available. To verify the correctness of the data, we run the clustered dataset through the Decision Tree (DT), Naive Bayes (NB), Multilayer Perceptron (MLP), and K-Nearest Neighbors (KNN) classification algorithms. We have used K-fold cross-validation [39,40] with the parameter *K* = 15. Figure 10 shows the graphical representation of correctly classified instances by DT, NB, MLP, and KNN classifiers.

From Figure 10, we can observe that the output of the correctly classified instance of the DT classifier is higher than that of other classifiers. So, we use the DT classifier on our cluster dataset for the final analysis. The results of our final correctly classified accuracy analysis are 87.3786%, as shown in Table 3 and Figure 11, through detailed accuracy by class and a confusion matrix of 103 instances. Precision measures the certainty of positive predictions. A value close to high (1) is considered a good precision score. Since the value of precision in Table 3 is almost close to one, we can conclude that our dataset has a good precision score. Similarly, as the recall score and F-measure score mentioned in Table 3 are close to high (1), we consider that the score of our dataset is very good.

### 4.6. Empirical Setup

This subsection discusses a practical consensus on how our work can be used in agriculture in the agricultural region of Bangladesh. If we consider Figure 12 and Figure 13 as a test area, we can see how we have applied IoT technology and how we are sending data to the application through that IoT technology. As we can see in Figure 12 and Figure 13, there is a control box with a Wi-Fi sensor connected to the NodeMCU, and with that control box, there are multiple wireless sensor networks, such as temperature, humidity, and soil moisture sensors, installed on agricultural land.

The NodeMCU is also connected to a water pump. The data from all these wireless sensors and the status of the water pump are sent to our cloud database via the NodeMCU API of the control box. The real-time data is shown to the farmers via the application from the cloud database. We selected a test area as a testbed that is 42.75 square feet in size (length 9.5 feet and width 4.5 feet) so that our structure may be used experimentally. We set up various sensors on the ground in this test area along with the control box, the specifics of which are displayed in Table 4. We transplanted 16 to 17 eggplant seedlings in our testbed and took the experimental data. Our testbed soil is a non-calcareous, dark gray floodplain soil. We experimentally collected 15 days of real-time data from the testbed in August and September 2020. When a vast region is involved, the parameters listed in Table 4 can be expanded by mesh connections. However, caution must be exercised to prevent rain and extremely hot or cold weather from harming the gadgets.

## 5. Discussion and Future Work

Our system framework is fundamentally centered around IoT, with IoT-enabled devices playing a crucial role in every aspect of our research work. Our proposed framework has been designed for farmers, and many useful features have been included. In addition, data analysis has been performed in a specific pattern through data mining techniques to make accurate predictions for farmers to plant and harvest crops by automatically analyzing the data. As a result, farmers can easily visualize the sensor data, make better use of plant management, and avoid wastage of both time and water due to automatic water management. Farmers will also receive daily and weekly weather forecasts based on the location of their agricultural land. As a result, by being vigilant in advance, the farmer can establish an appropriate strategy for harvesting the crop. Our application has a crop calendar with accurate information on 30 to 33 species of crops, detailing what farmers can do to plant, produce, harvest, and much more. As a result, they can use this calendar to work on planting or any other crop-related information without having to wait for the agricultural officer. The ability for farmers to save details about their planted crops in our program is one of the most intriguing aspects. As a result, the farmer will be informed automatically by our system during harvesting the crop. Basically, several physical layer properties and associated layer features have been covered in the proposed section of our research effort. Additionally, we demonstrate our proposed framework pattern for analysis through data mining in the performance analysis section. Although a good deal of the literature is focused on the agriculture monitoring system, there are not many automated agricultural monitoring, forecasting, or system architectures with several features that we incorporate in our system. Our framework is one of the most feature-rich monitoring systems for agriculture in Bangladesh. In addition, we created a new dataset [15] that will help researchers do further investigations from the perspective of Bangladesh agriculture in the future. We carried out our experiments on an experimental prototype based on Agriculture Statistics Bangladesh 2020 [5] data. Our experimental results can be applied in real-life situations to monitor agriculture.

One of the challenges in this work is to collect data on crop yield, planting, and season information in Bangladesh. More data can be collected in the future, and the data model and data analysis can be further improved. If the Message Queuing Telemetry Transport (MQTT) protocol can be added to our proposed system, it will become a more optimized system. However, we have introduced a cloud-based application instead of the MQTT protocol by providing a web-based application suitable for end users. In our framework, we have used a NodeMCU with an ESP8266 as the microcontroller unit in the control box, which is completely dependent on the Wi-Fi network. If we could use the mentioned microcontroller through the GPRS (General Packet Radio Service) network instead of another microcontroller, then it would be more useful to farmers in remote areas who do not have the technology to use smart phones or smart networks. While numerical data can be presented or transmitted without the integration of IoT devices, such a system may not effectively address real-world problems and provide comprehensive solutions. Additionally, the improvement of our physical gadgets’ outward structures is essential to keeping them functional and active for an extended period of time.

## 6. Conclusions

The proposed method holds the potential to benefit farmers, agricultural resource managers, researchers focusing on Bangladesh agriculture, and individuals interested in engaging with the country’s agricultural industry in the future. In this work, we collected data from various sources and created a dataset for our model. We analyzed the dataset in three steps: data pre-processing (data quality check and normalization), clustering to select an appropriate cluster dataset, and running various algorithms on the selected cluster for final data analysis. Our model achieved an accuracy of 87.3786%. Additionally, we included a crop informative calendar feature, providing farmers with crucial information about crop cultivation from sowing to harvesting. This work might inspire other researchers to explore and expand such a real-time monitoring system further. Such a system can help farmers predict the optimum time to plant their crops. Our work might bring impactful success to the agriculture environment in Bangladesh.

## Figures and Tables

**Figure 1 sensors-23-08472-f001:**
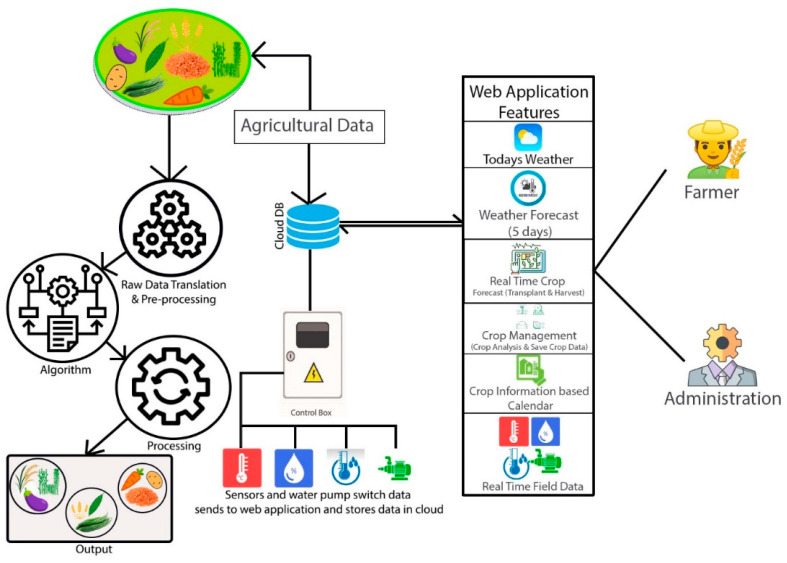
Overview of our proposed system.

**Figure 2 sensors-23-08472-f002:**
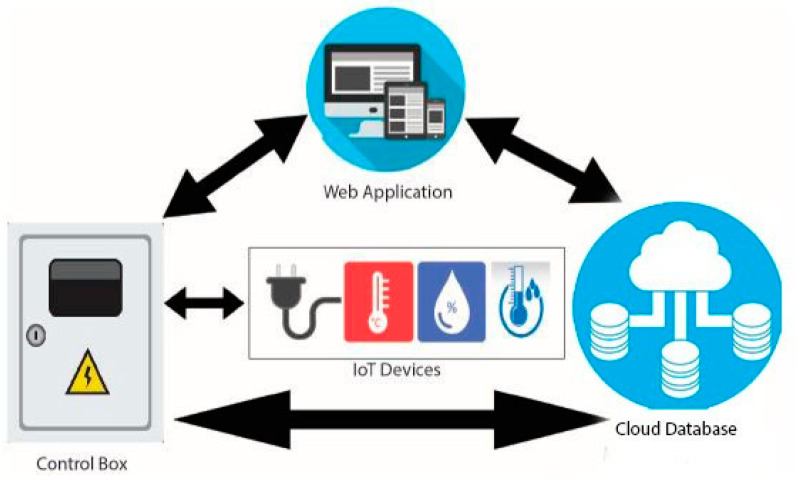
Overview of the system.

**Figure 3 sensors-23-08472-f003:**
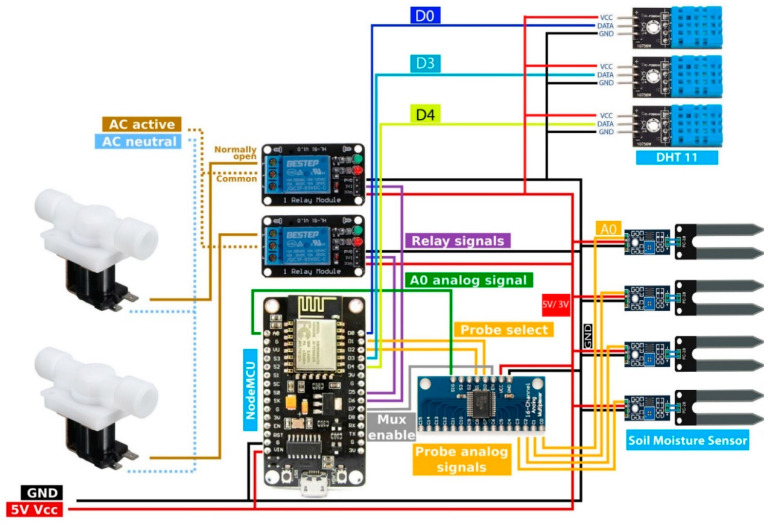
A block diagram of the circuit.

**Figure 4 sensors-23-08472-f004:**
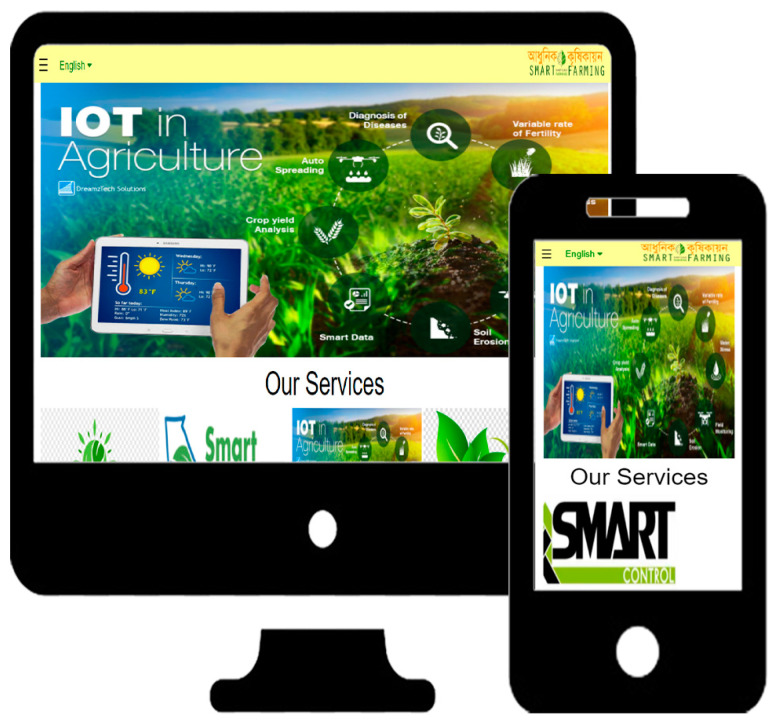
An overview of web and mobile applications.

**Figure 5 sensors-23-08472-f005:**
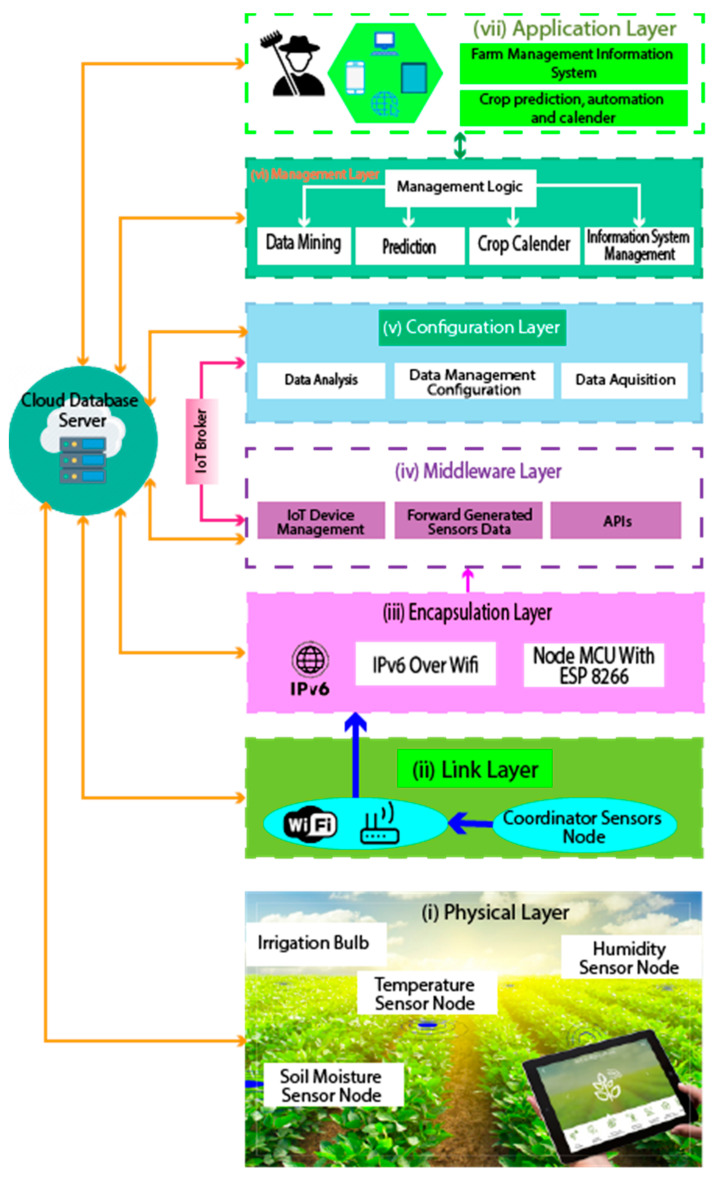
An architectural overview of the system.

**Figure 6 sensors-23-08472-f006:**
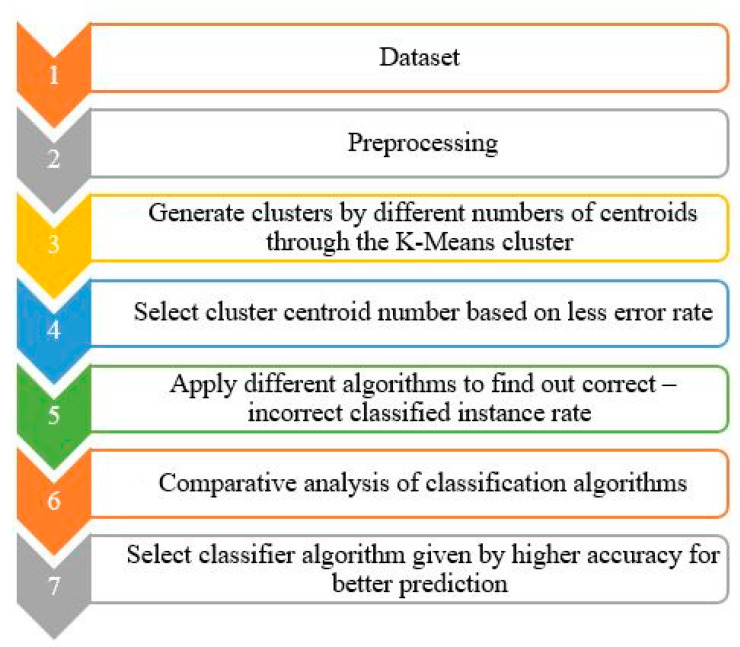
Flowchart of the data analysis step.

**Figure 7 sensors-23-08472-f007:**
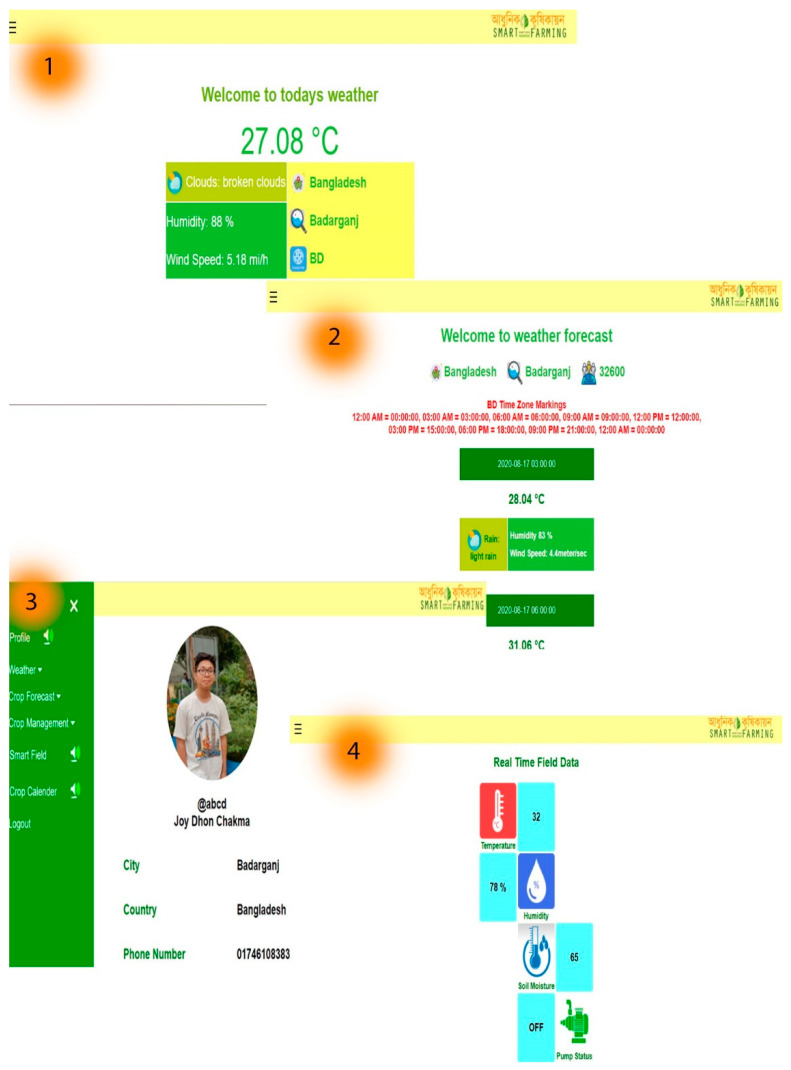
A cloud application that presents district-based (**1**) daily real-time weather, (**2**) weather forecasts up to 5 days, (**3**) farmers own profiles, and (**4**) real monitoring field data.

**Figure 8 sensors-23-08472-f008:**
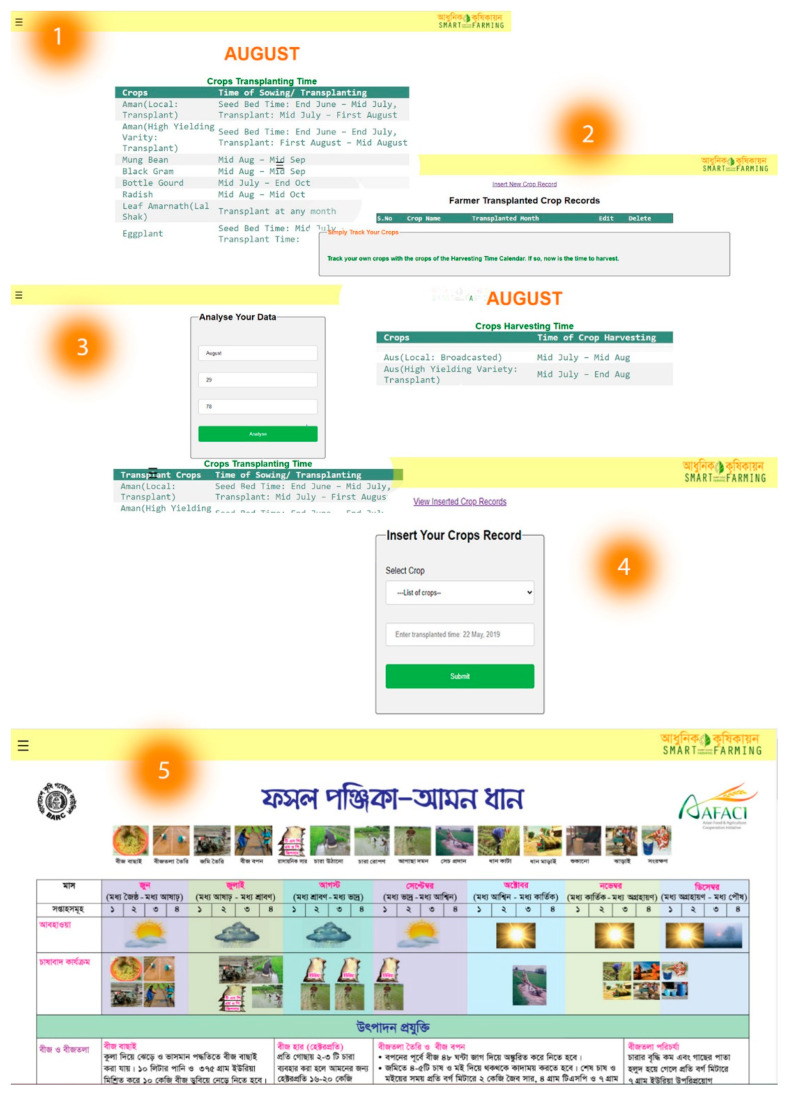
(**1**,**2**) real-time crop forecasts on planting and harvesting, (**3**) analyzing crop transplant and harvest time by inputting real-time data, (**4**) farmers can save their crop information, and (**5**) gathering knowledge from the crop calendar [32].

**Figure 9 sensors-23-08472-f009:**
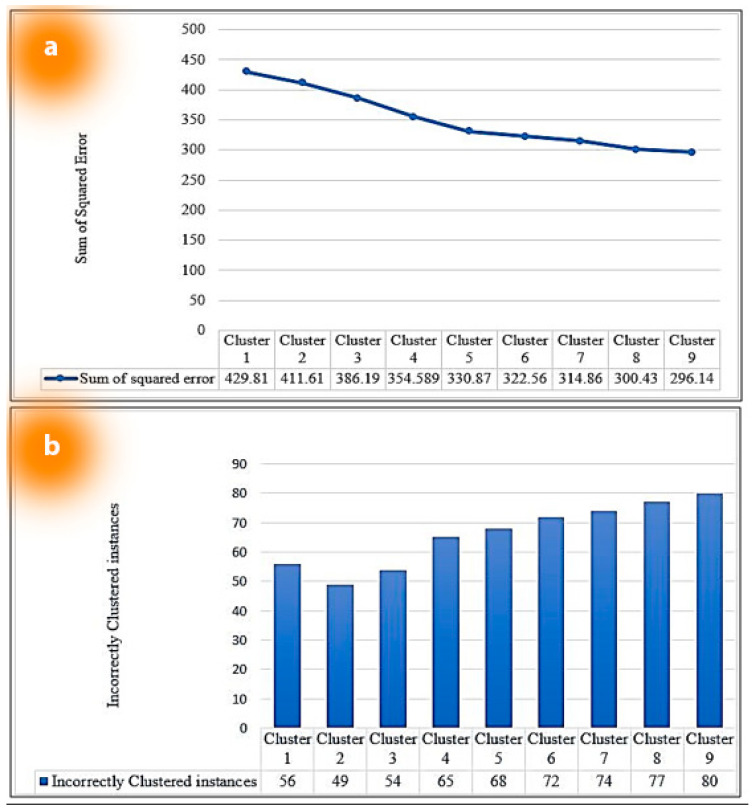
(**a**) Line chart of the SOSE of each cluster. (**b**) Clustered column chart of the ICI of each cluster.

**Figure 10 sensors-23-08472-f010:**
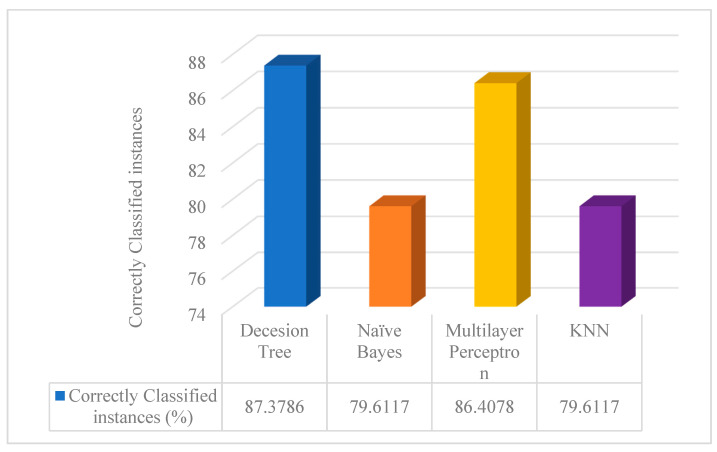
Accuracy comparison of DT, NB, MLP, and KNN classifiers.

**Figure 11 sensors-23-08472-f011:**
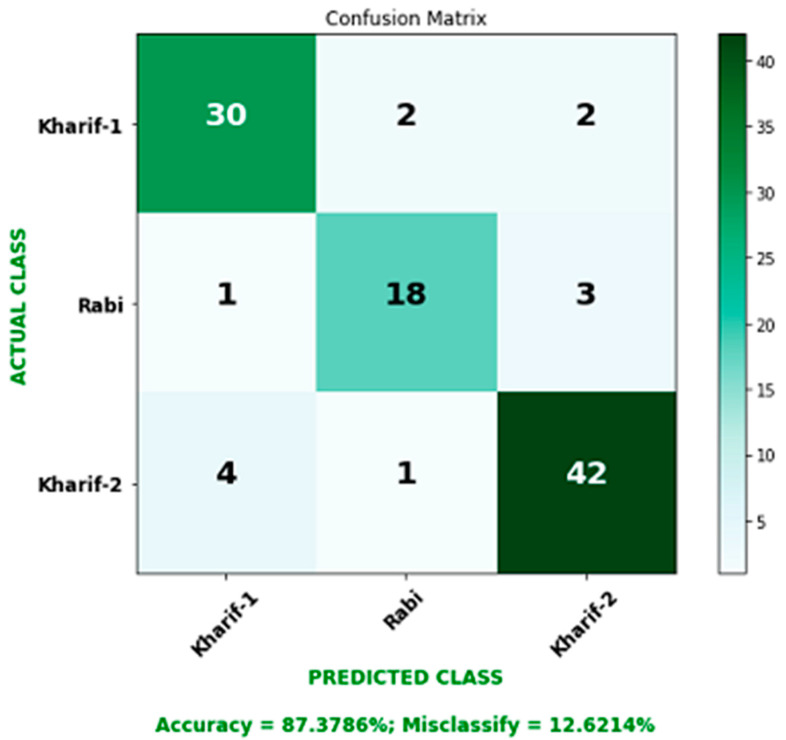
Accuracy results of DT classifiers.

**Figure 12 sensors-23-08472-f012:**
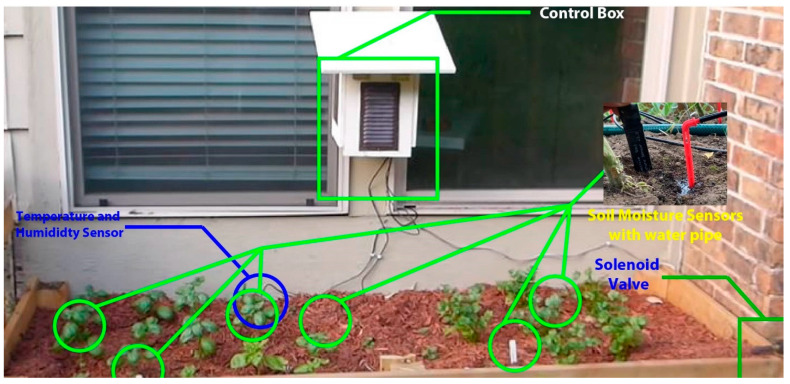
Real-time situation of a test area with a real example.

**Figure 13 sensors-23-08472-f013:**
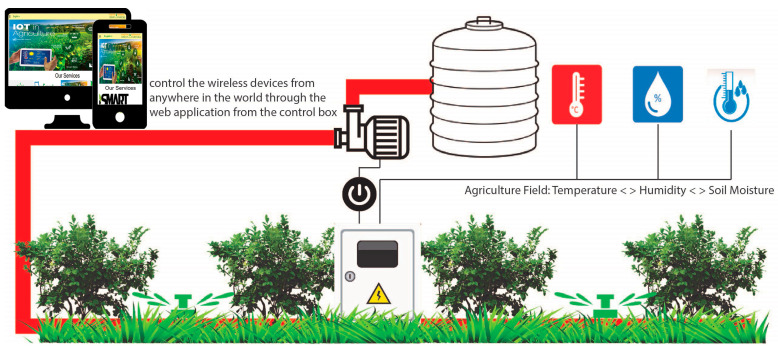
Real-time situation of a test area as a cartoon vector illustration.

**Table 1 sensors-23-08472-t001:** Comparison of our system framework features with other published research papers.

Feature	Our Proposed System Framework	S. Wolfert, L. Ge et al. [10] Article	Muangprathub, Jirapond et al. [11] Article	Liu, Shubo et al. [12] Article	Jash Doshi, Tirthkumar Patel et al. [13] Article
Web application	Yes	Yes	Yes	Yes	Yes
Mobile application	Yes	No	Yes	Yes	No
Real-time data collection	Yes	Yes	Yes	Yes	No
Real-time weather forecast	Yes	No	No	No	No
Weekly (5 days) weather forecast	Yes	No	No	No	No
Intelligent live system monitoring	Yes	Yes	Yes	Yes	Yes
Real-time data management	Yes	No	Yes	No	Yes
Crop data management	Yes	Yes	Yes	Yes	Yes
Manual data analysis	Yes	No	No	No	Yes
Live crop transplanting prediction on a month basis	Yes	No	No	No	No
Live crop harvesting prediction on a monthly basis	Yes	No	No	No	No
Crop transplanting to harvest time: an informative calendar	Yes	No	No	No	No

**Table 2 sensors-23-08472-t002:** SOSE and ICI results for each cluster.

Cluster Number	Cluster 1	Cluster 2	Cluster 3	Cluster 4	Cluster 5	Cluster 6	Cluster 7	Cluster 8	Cluster 9
SOSE	429.81	411.61	386.19	354.58	330.87	322.56	314.86	300.43	296.14
ICI	56	49	54	65	68	72	74	77	80

**Table 3 sensors-23-08472-t003:** Details of performance measurement matrices.

TP Rate	FP Rate	Precision	Recall	F-Measure	ROC Area	Class
0.971	0.087	0.846	0.971	0.904	0.974	Kharif 1
0.727	0.062	0.762	0.727	0.744	0.922	Rabi
0.894	0.018	0.977	0.894	0.933	0.971	Kharif 2

TP Rate = True Positive Rate, FP Rate = False Positive Rate, F-Measure = F1 Score or F Score, ROC Area = Receiver Operating Characteristic Area.

**Table 4 sensors-23-08472-t004:** Deployment parameters.

Parameters	Units	Remarks
NodeMCU	01	10 GPIO, every GPIO can be PWM with LUA Script
Relay Module	02	5 V DC
Solenoid Bulb	02	12 V
Probe Analog Signal	01	
DHT 11	01	3.3 V to 5 V
Sensing Probe	06	LM393 with 3.3 V to 5 V
Power Unit 01	01	3.3 V to 5 V
Power Unit 02	02	12 V
Wooden Box (Control Box)	01	11.5 inch ∗ 5.5 inch
Wire with male-female plug	01	H 2.54 mm

## Data Availability

On request, we might provide the data.

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
