# Peer review of "Smart Crop Cultivation System Using Automated Agriculture Monitoring Environment in the Context of Bangladesh Agriculture"

_sensors, 2023, doi:10.3390/s23208472_

Round 1

Reviewer 1 Report

Smart crop cultivation system using automated agriculture monitoring environment in the context of Bangladesh Agriculture is presented in this work and following are the observations and comments from this work that is to be addressed

1. This work seems to be novel from the country-specific but such works have been done across the globe for the past many years. So novelty perspective should be strongly illustrated.

2. Why is clustering used in this work? Is this about the unlabelled data collected?

3. Details of the attributes measured using sensors should be separately stated.

4. How forecasting is done? What will be forecasted?

5. Forecasting algorithm is novel?

6. How it is different from other works?

7. There are no comparison results with any state of the art of techniques.

8. There are plenty of works done with respect to IoT+cloud+forecating that can be compared and the result of a web application or a mobile application may not be a significant outcome of the research.

9. This work needs to be significantly improved from the research perspective.

Author Response

Below, we have addressed each of the reviewers' comments and have made the necessary revisions to enhance the quality and significance of our manuscript:

Reviewer 01 comments and our feedback:

Smart crop cultivation system using automated agriculture monitoring environment in the context of Bangladesh Agriculture is presented in this work and following are the observations and comments from this work that is to be addressed,

  1. This work seems to be novel from the country-specific but such works have been done across the globe for the past many years. So novelty perspective should be strongly illustrated.

Response:

“The characteristics developed here are country-specific innovations, and many other regions of the world already use them. Nobody has yet to come across a scientific article that has all these traits. Additionally, we have talked about the analysis of datasets that are country-specific. The primary contributions listed in the introduction are also covered in detail.”

  1. Why is clustering used in this work? Is this about the unlabeled data collected?

Response:

“We have used cluster in this work because initially when we collected the data and collected a dataset, the data was very unlabeled and had similar characteristics data. As a result we do clustering for preprocessing. This process of clustering is described in two broad parts. The first part is covered in Sub-Section 3.2.3 (II) and the second part is mentioned first in Section 4.4 Data Curation and Results.”

  1. Details of the attributes measured using sensors should be separately stated.

Response:

“In Sub Section 3.2. System Implementation, 3.2.1. Crops data generation, real-time forecasting, and empirical setup section, the properties measured by the sensors are covered in depth.”

  1. How forecasting is done? What will be forecasted?

Response:

“In order to determine which crops can be planted and which crops can be harvested in the continuing monthly cycle, our system architecture gathers real-time data from sensors and analyzes it alongside historical data. Additionally, by integrating APIs, the area where the farmer's cropland is located can predict the weather in real time for the next five days.”

  1. Forecasting algorithm is novel?

Response:

“Undoubtedly it is novel.”

  1. How it is different from other works?

Response:

“Numerous nations currently and in the future carry out this type of work. There is no confirmation that a farmer will have access to all the characteristics in one application, even though more or less similar features have been observed in any study conducted thus far. However, our system structure is flexible and packed with features. It must be merged with the country's statistical historical data in order to be used in the agriculture sector of any country. However, South East Asian nations may easily implement our system framework.”

  1. There are no comparison results with any state of the art of techniques.

Response:

“We don't think that it is necessary to provide separate comparative results for the techniques others used and what we used for data analysis and the programs utilized as input to the devices. It's also unclear whether or not it will fit with our paper. Also, if we were considering this paper as a case study or a review paper, that table or result might be required.”

  1. There are plenty of works done with respect to IoT + cloud + forecasting that can be compared and the result of a web application or a mobile application may not be a significant outcome of the research.

Response:

“We already did it in Table 01.”

  1. This work needs to be significantly improved from the research perspective.

Response:

“We have made a lot of effort to improve our job. In the revolutionary IoT sector, fresh research will be conducted continuously to advance technology. The touch of better technology will be accessible when the price of IoT industry devices increases. However, we have made an effort to provide as many functions as we can in a web or mobile application while keeping below a realistic cost, keeping farmers in mind. We wrote this research paper to address issues such as how to automate farming, forecast crops and weather, monitor farm equipment in real time, and show how a farmer can use our framework to develop opportunities for producing eco-friendly crops without the assistance of an agricultural officer. Many things can be integrated if desired. However, we have tried to squeeze all the IoT devices available in this region and as many resources as we can explore. There will be more work on this in the future. But as of now, no one can see such a numerical feature. We strongly hope that people who work on it in the future will find our paper to be among the most pertinent ones.”

Reviewer 02 comment and our side explanation:

Comment: Avoid general sentences. E.g., ‘The Internet of Things (IoT) is one of the most progressive ideas in the modern era.’ A lot of sentences are poorly constructed. E.g., ‘Bangladesh is an agro-based country and its economy largely depends on it’ – what is ‘it’? ‘is primarily dependent on this agricultural sector’ – what agricultural sector?

Response:

“As per reviewer suggestion we have tried to rewrite this type of poorly constructed sentences.”

Comment: ‘It is said that the land of Bangladesh is fertile and it seems that the farmers are cultivating gold’ – this is not scientific: ‘it is said’, ‘are cultivating gold’. ‘to IoT technology’ – any acronym must be first written in full.

Response:

“As per the reviewer's suggestion we have changed this sentence. The full form of IoT has been written many times in Abstract, so we are not writing its full form everywhere.”

Comment: Sentences are full of word repetitions. E.g., ‘in the agriculture sector has been carried out to develop agricultural benefits and smart agriculture infrastructure’. ‘The application of IoT technology in agriculture has brought great revolutionary changes in the agricultural environment by focusing on multiple challenges and examining different complexities’ – zero info.

Response:

“The first phrase discusses the use of technology to enhance agricultural infrastructure. On the other hand, the second statement clearly discusses the revolutionary IoT technology. And we believe that the meanings of these statements vary on their own.”

Comment: ‘We now hope that with the advancement of this technology, using IoT technology’, ‘Advanced technology is not always, but most of the time benefits people’ – poorly constructed.

Response:

“We have added one more sentence with it to understand why we actually wrote the line, “Advanced technology is not always, but most of the time benefits people”. On the other hand this statement, in our opinion, is appropriate and goes well with the introduction.”

Comment: A lot of ideas are not substantiated. E.g., ‘the IoT has been able to play a major role in everyday life by increasing our perceptions and skills to change the environment around us. IoT is applied in both diagnostics and control, especially in the agro-industrial areas and environmental sectors. It also helps the farmer to provide information about the source and characteristics of the grain or product.’

Response:

“These sentences, in our opinion, are appropriate and reliable. All of the technology we use today has an IoT component. The Internet of Things touches everything, from checking the weather on your phone to using fingerprint sensors, lights, and remotes to operate fans, switches, and motors to pump your water!”

Comment: ‘Later, by analyzing the data in the cloud, various types of forecasts are made available to the farmers’ – ‘later’ when?

Response:

“We have changed the sentence. While changing the sentence, the meaning remained unchanged.”

Comment: ‘how the combination of machine learning, big data, and IoT networks has had a profound effect on farms and agriculture in the Old Testament’ – just checked the paper, it does not refer to the Old Testament. It is strange to relate verifiable data to the Old Testament anyway.

Response:

“We have changed the review of ours thanks to reviewer.”

Comment: The in-text citation system is unusual and not uniform. A lot of cited sources are not developed. ‘They created a control system based on data management and node sensors’ – they who? ‘Their method was put into place to regulate’ – their who?

Response:

“They mean article 8 which is mentioned in the previous sentence. They mean article 9 which is mentioned in the previous sentence. We can't constantly refer to these authors by their names in every phrase in the same way. It can ruin a paper's aesthetic appeal. Each linked work that we have reviewed has been presented in an effort to be unique.”

Comment: ‘A survey of the literature that was centered on studies and analyses of the application of IoT in modern farming’ – incomplete sentence.

Response:

“Don't make a decision based solely on the reference sentence, please. The sentences that follow after all contain the same meaning.”

Comment: ‘Their research and analysis showed how China’ – their who?

Response:

“They mean article 11 which is mentioned in the 2nd next sentence. We can't constantly refer to these authors by their names in every phrase in the same way. It can ruin a paper's aesthetic appeal. Each linked work that we have reviewed has been presented in an effort to be unique.”

Comment: The manuscript is full of such instances. ‘Thebasy overviewed’ – Thebasy?

Response:

“Since no one questioned this word, it went unnoticed. Actually, the error was in the typing. We appreciate the review.”

Comment: The references are not in order and some of them seem to be missing in the text.

  1. ‘In Ref. [17] article authors presented’, ‘In Ref. [32] article authors discusses’ (‘discuss’ anyway) – very unusual.
  2. ‘A literature review on the role of Internet of Things technologies in agriculture that explored the varied effects of IoT in agriculture’ – poorly constructed.
  3. Figures and tables should be improved, unified as style, and thoroughly explained. More development and depth of the methodology and analysis are needed. The reference list is not properly edited.

Response:

 “As suggested by the reviewer, we made an effort to sort the reference numbers. The design and specifics of the figures and tables have been improved.”

Comment: The relationship between cloud-based IoT data analytics and digital twins in smart farming as regards smart crop cultivation systems using automated agriculture monitoring environment  has not been covered, and thus such sources can be cited:

  1. Nica, E.; Popescu, G.H.; Poliak, M.; Kliestik, T.; Sabie, O.-M. Digital Twin Simulation Tools, Spatial Cognition Algorithms, and Multi-Sensor Fusion Technology in Sustainable Urban Governance Networks. Mathematics 2023, 11, 1981. https://doi.org/10.3390/math11091981

  2. Pop, R.A., Dabija, D.C., Pelau, C., Dinu, V. 2022. Usage Intentions, Attitudes, and Behaviours towards Energy-Efficient Applications during the COVID-19 Pandemic. Journal of Business Economics and Management, 23(3), pp.668-689. https://doi.org/10.3846/jbem/2022/16959

  3. Andrei, J. V., Popescu, G. H., Nica, E., & Chivu, L. (2020). The impact of agricultural performance on foreign trade concentration and competitiveness: empirical evidence from Romanian agriculture. Journal of Business Economics and Management, 21(2), 317-343. https://doi.org/10.3846/jbem.2020.11988

Response:

“In our study, we cited and briefly discussed the first reference. We thought it should be noted because it is somewhat similar to our scope. And we apologize for the others; we did not cite them because there was no fit with our scope.”

Comments on the Quality of English Language

A lot of sentences are poorly constructed. E.g., ‘Bangladesh is an agro-based country and its economy largely depends on it’ – what is ‘it’? ‘is primarily dependent on this agricultural sector’ – what agricultural sector? ‘We now hope that with the advancement of this technology, using IoT technology’, ‘Advanced technology is not always, but most of the time benefits people’ – poorly constructed. ‘They created a control system based on data management and node sensors’ – they who? ‘Their method was put into place to regulate’ – their who? ‘A survey of the literature that was centered on studies and analyses of the application of IoT in modern farming’ – incomplete sentence. ‘Their research and analysis showed on how China’ – their who? The manuscript is full of such instances. ‘Thebasy overviewed’ – Thebasy? ‘A literature review on the role of Internet of Things technologies in agriculture that explored the varied effects of IoT in agriculture’ – poorly constructed.

Response:

“We have tried to improve the quality of the English language according to the reviewer's words.”

Reviewer 03 comments and our revisions:

Manuscript

Title: „Smart crop cultivation system using automated agriculture monitoring environment in the context of Bangladesh Agriculture”

Authors: Md. Bayazid Rahman , Joy Dhon Chakma, Dr. Abdul Momin , Dr. Shahidul Islam , Dr. Md Ashraf Uddin , Dr. Md Aminul Islam

Dear Authors

I revised the manuscript: "Smart crop cultivation system using automated agriculture monitoring environment in the context of Bangladesh Agriculture” submitted to the „Sensors” Journal. The paper is very interesting. However, I have some concerns, which need to be addressed.

Line 2-4. Article topic

The theme of the article is concise and accurately reflects the content of the article.

The structure of the article, divided into chapters and subchapters, is clear and logical.

It is noticeable that there are a lack of keywords in the topic of the article, especially: ”prediction, data mining, internet of things etc”.

This means the broader context of the research findings. In my opinion, however, the content of the article is a form of case study. Please consider converting the article topic to better represent the content of the article.

Response:

“As per the reviewer's suggestion, the issues that we have worked on in the research have been changed and updated in the keyword section. Lines 24-25.”

Abstract

Line 13-23. The content of the abstract should indicate the measurable effect of the research in terms of giving a numerical value and indicating the most important conclusions generalised even within the implementation of the case study. Please take this into account.

The abstract is a self-contained part of the article which requires repetition of any explanatory notes.

The authors suggest the goal and scope of the work, but the main goal of the work is split between individual specific tasks, which are shown implicitly as sub-results. The descriptions of the results are very abbreviated.

Line 20-21. „….several experiments are adopted to check the….” The expression is imprecise and should have no place in the abstract.

It is difficult to identify the leading result and the leading conclusion of the research. Please take this into account.

Response:

“We follow the reviewer's suggestion „….several experiments are adopted to check the….” this sentence is omitted. Also, we have tried to highlight our work as much as possible in the abstract with leading results and conclusions. For which we try to make changes according to the suggestions of the reviewer in the abstract.”

Keywords. Line 24-25. The authors used the full spectrum of matching wording. Keywords contain word clusters and are too literal, but represents the spectrum of information well. The order of keywords should follow the concept of „from generalities to specifics”. Keywords may also reflect the order in which the research issues are addressed.  Please consider changing the order of keywords.

Please try to arrange your keywords according to a coherent concept and dispense with near-meaningful expressions and duplicate messages.

Response:

“As per the reviewer suggestion, the issues that we have worked on in the research have been changed and updated in the keyword section. Lines 24-25.”

  1. Introduction

The state of the knowledge presented is relevant to the goal and scope of the research. The presentation of the state of knowledge in the chapter is quite general and its form is quite brief. The authors have used only four literature sources to outline the state of research in the spectrum discussed. Please consider elaborating the content provided in more detail in the context of an attractive and internationally recognized research topic.

Response:

“As per reviewer's suggestion we have added more references in “Introduction” Chapter.”

Line 65-69. The included information indicates research results related to a case study analysis (case study), with limited possibilities of generalization. This is a conclusion and should conclude the analysis of the research results instead making a form of hypothesis.

Response:

“As per reviewer's suggestion we have added few more details.”

Line 70 – 76. A formal presentation of the content division of the article is not necessary. Please take this into consideration.

Response:

“Based on the other reference articles format style, we decided to mention the arrangement of article for clarity of the readers.”

  1. Related Work

The lack of an indication of the goal and scope of the research in Chapter 1, even in a descriptive and general way, makes it difficult for the reader to understand the intention of the content introduced in Chapter 2. The reader has the impression of knowing rather general and difficult to relate thematic content. A further analysis of the state of the knowledge is continued in the next chapter. This division is not necessary.

The order of citation of the literature sources coinciding with the ordinal numbers is disrupted because literature source with ordinal number [5] only appear in line 233. Please discuss this with the journal editors or correct the problem.

Response:

“As per reviewer suggestion we shuffle and corrected the order of the sources number according to sensor journal format.”

The order of citation of the literature sources coinciding with the ordinal numbers is disrupted because literature source with ordinal number [15] only appear in line 414. Please discuss this with the journal editors or correct the problem.

Response:

“As per reviewer suggestion we shuffle and corrected the order of the sources number according to sensor journal format.”

The order of citation of the literature sources coinciding with the ordinal numbers is disrupted because literature items with ordinal numbers [24] to [31] only appear in lines: Line 403 [24], Line 368 [25], Line 233 [31] . Please discuss this with the journal editors or correct the problem.

Response: “As per reviewer suggestion we shuffle and corrected the order of the sources number according to sensor journal format.”

The literature source number [26] was not self-quoted in the text of the article in the correct order. Please correct this. (only then in Line 384)

Response: “As per reviewer suggestion we corrected to the standard MDPI Sensors format.”

The literature source number [27] was not self-quoted in the text of the article in the correct order. Please correct this. (only then in Line 384)

Response: “As per reviewer suggestion we corrected to the standard MDPI Sensors format.”

The literature source number [28] was not self-quoted in the text of the article in the correct order. Please correct this. (only then in Line 384)

Response: “As per reviewer suggestion we corrected to the standard MDPI Sensors format.”

The literature source number [29] was not self-quoted in the text of the article in the correct order. Please correct this. (only then in Line 451)

Response: “As per reviewer suggestion we corrected to the standard MDPI Sensors format.”

The literature source number [30] was not self-quoted in the text of the article in the correct order. Please correct this. (only then in Line 451)

Response: “As per reviewer suggestion we corrected to the standard MDPI Sensors format.”

Line 80. „….Andreas et al. [7] provided a literature review demonstrating….” We do not refer to a 'literature review' but to 'research results'. Please take this into account.

Response: “As per reviewer suggestion we have changed it from “Literature review” to “study of research”.” 

Line 97. „….RFID…” Please explain the introduced abbreviations of terms, possibly as soon as they are introduced in the content of the article. Even popular acronym names need to be explained in scientific articles.  A single explanation is sufficient in the execution of the explanation.

Response:

“As per reviewer suggestion we have this abbreviation to full form.”

Line 100. „….As surveyed in [13]….”This formulation is too simplistic. A ordinal number cannot be implicitly considered as a source of knowledge. The sentence should be completed with the words: scientific article.

Response:

“As per reviewer suggestion we have changed it into “scientific article”.”

Line 103 – 122. Research results or scopes of research of analyzed scientific articles should be cited in the context of convergence with the own research task. In this sense, the fragment of chapter two and many other parts are too general and weakly linked to the goal and scope of your own research.  Please correct this.

Response:

“We remove a citation from our paper. We believe that the referenced work is considerably distinct from our scope and goals, as suggested by the reviewer.

The removed paper:

RL, Raghavi, and A. Umamageswari. "Modern Irrigation based on Web Weather Forecast." (2018).”

Line 146. „….The AgriTrust….” Please explain the introduced abbreviations of terms, possibly as soon as they are introduced in the content of the article. Even popular acronym names need to be explained in scientific articles.  A single explanation is sufficient in the execution of the explanation.

Response:

“As per reviewer suggestion we added the explanation of abbreviation “AgriTrust”.”

Line 160. Table 1. Comparative features and criteria appear too late for understanding the intention and tasks of the presented scientific article. Please introduce the criteria and scopes of your own research earlier in the content of the article.

Lack of reference to 'Table 1' in the content of chapter two. Please correct this if Table 1 should stay in the content of Chapter Two.

Response:

“Thank you for noticing it. We move the table in a suitable chapter where it actually belong. We move it to sub section 3.2. (System Implementation).”

  1. Our Empirical Approach

Line 175-178. „…The purpose of this work is to send data from the farm to the web application or smart phone through the data management controller named NodeMCU. The design and implementation overview of this system is divided into 3 components namely hardware, web/mobile application, and cloud database, as shown in Figure 2…. ”

There is a noticeable dissonance between: the theoretical, informative preparation of Chapters 1 and 2, the research methods, the verification content of Chapter 4, and the notation from lines 175 to 178 of Chapter 3. The highlighted section of the chapter contains a mental shortcut that requires two actions: a detailed explanation of the goal and scope of the research and the creation of research methods and results logically connected to the stated goal and scope of the research. It should also be noted about the order of presentation of the research results, which depends on the logical sequence of the appearance of the sub-tasks and the support of the research methods and research stands. Please take this into account in the processes of organising the content of the scientific article.

Response:

“According to the reviewer's point of view, we have tried to cover the inconsistency of the theoretical and informative preparation in the first paragraph of Chapter 3.1. We have added some new information to this paragraph to resolve this inconsistency.”

Line 176. „….NodeMCU….” Please explain the introduced abbreviations of terms, possibly as soon as they are introduced in the content of the article. Even popular acronym names need to be explained in scientific articles.  A single explanation is sufficient in the execution of the explanation.

Response:

“As per reviewer suggestion we added the explanation of abbreviation “NodeMCU” and “ESP8266”.”

Line 183, 220, 249, 488, . „….DHT 11….” Please explain the introduced abbreviations of terms, possibly as soon as they are introduced in the content of the article. Even popular acronym names need to be explained in scientific articles.  A single explanation is sufficient in the execution of the explanation.

Response:

“We have added the explanation of “what DHT 11 device is.” in a single explanation in our paper. We have only added the explanation to the first abbreviation. Abbreviations are left in the rest only. Since the explanation has been done in the very first one, we think that it does not need an explanation in the latter ones.”

Line 183. „….a node MCU…” Please explain the introduced abbreviations of terms, possibly as soon as they are introduced in the content of the article. Even popular acronym names need to be explained in scientific articles.  A single explanation is sufficient in the execution of the explanation.

Response:

“We have added the explanation of NodeMCU in chapter 3.1 as per suggestion of reviewer. Since the explanation has been done in the very first one, we think that it does not need an explanation in the latter ones.”

Line 192. „….MIT…..” Please explain the introduced abbreviations of terms, possibly as soon as they are introduced in the content of the article. Even popular acronym names need to be explained in scientific articles.  A single explanation is sufficient in the execution of the explanation.

Response:

“We have added the explanation of “what MIT app inventor is.””

Line 196. „….API….” Please explain the introduced abbreviations of terms, possibly as soon as they are introduced in the content of the article. Even popular acronym names need to be explained in scientific articles.  A single explanation is sufficient in the execution of the explanation.

Response:

“We have added the full form of “API.””

Line 204. „…Wi-Fi network…” Please explain the introduced abbreviations of terms, possibly as soon as they are introduced in the content of the article. Even popular acronym names need to be explained in scientific articles.  A single explanation is sufficient in the execution of the explanation. For example: (wireless fidelity)

Response:

“We have added the full form of “Wi-Fi””

Line 206. „…IPv6….” Please explain the introduced abbreviations of terms, possibly as soon as they are introduced in the content of the article. Even popular acronym names need to be explained in scientific articles.  A single explanation is sufficient in the execution of the explanation.

Response:

“We have added the full form of “IPv6””

Line 201 - 215: "...(i), (ii), (iii), (iv), (v), (vi), (vii)...". Please indicate the proposed layer identifiers as designations in Figure 5. The notation is incomprehensible. Please correct it.

Response:

“We updated figure 5 as per reviewer suggestion.”

Line 221. „….ESP-8266 module….” Please explain the introduced abbreviations of terms, possibly as soon as they are introduced in the content of the article. Even popular acronym names need to be explained in scientific articles.  A single explanation is sufficient in the execution of the explanation.

Response:

“We already explained it before in chapter 3.1 as per reviewer suggestion.”

Line 237. „…ESP8266 Node MCU…” The notation of the same acronym designations will change in the body of the article. Please standardise the notation of names and acronyms.

Response:

“We already explained it before in chapter 3.1 as per reviewer suggestion.”

Line 369. „…(i)….” Why do the authors not number the mathematical formulae with numerical designations? There is an apparent conflict with the description markings of Figure 5.

Response:

“We have changed the mathematical formulae numerical to number designation.”

Line 385. „….(ii)…..” Why do the authors not number the mathematical formulae with numerical designations? There is an apparent conflict with the description markings of Figure 5.

Response:

“We have changed the mathematical formulae numerical to number designation.”

Line 391. Figure 7. The content of the textual information in the figure is unreadable. Please correct this.

Response:

“We have changed the figure and try to make textual information in the figure is readable.”

Line 394. Figure 8. There is a lack of reference in the body of the article to Figure 8, in the assigned chapter to Figure 8. Please complete this.

The content of the textual information in the figure is unreadable. Please correct this.

Response:

“We have changed the figure and try to make textual information in the figure is readable. And also add a reference for this that we forgot to add in this figure 8 (5 no.). Thank you for that.”

  1. Experimental Setup and result discussion

The assumptions presented for the study material and research methods, when confronted with the parameter assumptions in Table 1, are difficult to harmonise and reconcile. There is a lack of clear separation between the scope of the research and the coinciding results.

Response:

“Our study and research work is heavily defends on the application layer because of IoT works and physical layer. And both comes together perfectly with the associated layers. And from the associated layer one of the important layer to get accuracy and also show the correct result to farmers is the configuration layer where we did data analysis, data acquisition and data management configuration. In table 1 the features are came from application layer. Actually when we think about farmers is that actually how much feature we are giving them. So that we compare other scientific paper application layers feature with our application layer feature. As per reviewer suggestion we moved this table 1 from chapter 2 and placed it in chapter 3.2.”

Line 429. „…SOSE….” Sum of the squared errors, SSE, is defined as follows: SSE = N ∑ ei2 =∑ (xi − ˆxi)2. Is this the same notation as the SOSE parameter? There is no explicit definition of the SOSE parameter in the form of a mathematical model.

Response:

“As per reviewer suggestion we explained the SOSE parameter with a mathematical model.”

Line 429. „…. incorrectly clustered instances (ICI)….” There is a lack of an explicit definition of the ICI parameter in the form of a mathematical model.

Response:

“As per reviewer suggestion we explained the ICI parameter with a mathematical model.”

Line 434. „…. ISI ….” Please explain the introduced abbreviations of terms, possibly as soon as they are introduced in the content of the article. Even popular acronym names need to be explained in scientific articles.  A single explanation is sufficient in the execution of the explanation.

Response:

“We already have the explanation of ICI in cluster section as per reviewer suggestion.”

Line 442. Figure 9. Lack of markings (a) and (b) on the individual graphics of Figure 9.

Response:

“Thanks to reviewer we updated the figure 9.”

Line 457. „….Table 03…” This is a multiplication of the synonyms of Table 3. Please homogenise the notation. Please do not create alternative designations for the same element.

Response:

“Thanks to reviewer we updated the designations of table 3 and other notations.”

Line 463. Table 3. „…TP Rate, FP Rate, Precision, Recall, F-Measure, ROC Area,….” Please explain the acronyms introduced in the table independently of the content of the chapter.

Response:

“We add each of the acronyms full form in Table 3.”

  1. Discussion & Future Work

Line 512, 514. „…MQTT….” Please explain the introduced abbreviations of terms, possibly as soon as they are introduced in the content of the article. Even popular acronym names need to be explained in scientific articles.  A single explanation is sufficient in the execution of the explanation.

Response:

“We added the acronym MQTT full form.”

Line 492 - 509 The content fits more with Chapters 1 and 2 and should supply the goal and scope of the work. Please take this into account.

Response:

“As per reviewer suggestion we move the first 4 to 5 sentence in chapter 1 introduction. And we added there a valuable discussion about our work.”

Line 510-521. „….Our main challenge in this work is to collect data on crop yield, planting and season nformation in Bangladesh….”This formulation reminds the scope of the study but is not articulated earlier. It is useful to organise the content of the article and create a logical sequence of statements, results and conclusions. Please take this into account.

Response:

“We have not varied much in our challenge. We particularly believe that our challenges are valid and that the counterarguments to the challenges are valid.”

The content presented in Chapter 5 does not constitute a valuable discussion of the results because, as in Chapter 4, the range of comparative research findings cited from the analysis of the state of knowledge is insufficient even for a so-called case study. Please complete and expand the discussion of the results.

Response:

“We regret that Chapter 5 lacks constructive discussion. We have tried to make the whole discussion part different according to the reviewer's suggestion and have done many constructive discussions. Hope the reviewer will enjoy reading it.”

  1. Conclusion

Line 523 - 530. The content of the chapter matches the opening of the discussion about results rather than the conclusion.

Line 531 – 539. The second part of Chapter 6 better represents the concept of the conclusion.

The presentation of results in this second part of the chapter is adequate to present the conclusions. Please take this into account.

Response:

“As per reviewer comments, we removed the contents that matches with other chapters content. And also we kept only second part as per reviewer suggestion.”

References

Line 540-624. Please check for literature sources available online. Please verify whether other forms of availability and other forms of publisher addresses for online publication sources can be indicated.

Response:

“We investigated, and their availability is consistent.”

Reviewer 2 Report

Avoid general sentences. E.g., ‘The Internet of Things (IoT) is one of the most progressive ideas in the modern era.’ A lot of sentences are poorly constructed. E.g., ‘Bangladesh is an agro-based country and its economy largely depends on it’ – what is ‘it’? ‘is primarily dependent on this agricultural sector’ – what agricultural sector? ‘It is said that the land of Bangladesh is fertile and it seems that the farmers are cultivating gold’ – this is not scientific: ‘it is said’, ‘are cultivating gold’. ‘to IoT technology’ – any acronym must be first written in full. Sentences are full of word repetitions. E.g., ‘in the agriculture sector has been carried out to develop agricultural benefits and smart agriculture infrastructure’. ‘The application of IoT technology in agriculture has brought great revolutionary changes in the agricultural environment by focusing on multiple challenges and examining different complexities’ – zero info. ‘We now hope that with the advancement of this technology, using IoT technology’, ‘Advanced technology is not always, but most of the time benefits people’ – poorly constructed. A lot of ideas are not substantiated. E.g., ‘the IoT has been able to play a major role in everyday life by increasing our perceptions and skills to change the environment around us. IoT is applied in both diagnostics and control, especially in the agro-industrial areas and environmental sectors. It also helps the farmer to provide information about the source and characteristics of the grain or product.’ ‘Later, by analyzing the data in the cloud, various types of forecasts are made available to the farmers’ – ‘later’ when? ‘how the combination of machine learning, big data, and IoT networks has had a profound effect on farms and agriculture in the Old Testament’ – just checked the paper, it does not refer to the Old Testament. It is strange to relate verifiable data to the Old Testament anyway. The in-text citation system is unusual and not uniform. A lot of cited sources are not developed. ‘They created a control system based on data management and node sensors’ – they who? ‘Their method was put into place to regulate’ – their who? ‘A survey of the literature that was centered on studies and analyses of the application of IoT in modern farming’ – incomplete sentence. ‘Their research and analysis showed on how China’ – their who? The manuscript is full of such instances. ‘Thebasy overviewed’ – Thebasy? The references are not in order and some of them seem to be missing in the text. ‘In Ref. [17] article authors presented’, ‘In Ref. [32] article authors discusses’ (‘discuss’ anyway) – very unusual. ‘A literature review on the role of Internet of Things technologies in agriculture that explored the varied effects of IoT in agriculture’ – poorly constructed. Figures and tables should be improved, unified as style, and thoroughly explained. More development and depth of the methodology and analysis are needed. The reference list is not properly edited.
The relationship between cloud-based IoT data analytics and digital twins in smart farming as regards smart crop cultivation systems using automated agriculture monitoring environment  has not been covered, and thus such sources can be cited:
Nica, E.; Popescu, G.H.; Poliak, M.; Kliestik, T.; Sabie, O.-M. Digital Twin Simulation Tools, Spatial Cognition Algorithms, and Multi-Sensor Fusion Technology in Sustainable Urban Governance Networks. Mathematics 2023, 11, 1981. https://doi.org/10.3390/math11091981
Pop, R.A., Dabija, D.C., Pelau, C., Dinu, V. 2022. Usage Intentions, Attitudes, and Behaviours towards Energy-Efficient Applications during the COVID-19 Pandemic. Journal of Business Economics and Management, 23(3), pp.668-689. https://doi.org/10.3846/jbem/2022/16959
Andrei, J. V., Popescu, G. H., Nica, E., & Chivu, L. (2020). The impact of agricultural performance on foreign trade concentration and competitiveness: empirical evidence from Romanian agriculture. Journal of Business Economics and Management, 21(2), 317-343. https://doi.org/10.3846/jbem.2020.11988

A lot of sentences are poorly constructed. E.g., ‘Bangladesh is an agro-based country and its economy largely depends on it’ – what is ‘it’? ‘is primarily dependent on this agricultural sector’ – what agricultural sector? ‘We now hope that with the advancement of this technology, using IoT technology’, ‘Advanced technology is not always, but most of the time benefits people’ – poorly constructed. ‘They created a control system based on data management and node sensors’ – they who? ‘Their method was put into place to regulate’ – their who? ‘A survey of the literature that was centered on studies and analyses of the application of IoT in modern farming’ – incomplete sentence. ‘Their research and analysis showed on how China’ – their who? The manuscript is full of such instances. ‘Thebasy overviewed’ – Thebasy? ‘A literature review on the role of Internet of Things technologies in agriculture that explored the varied effects of IoT in agriculture’ – poorly constructed.

Author Response

The Reviewer's comment and our side explanation:

Comment: Avoid general sentences. E.g., ‘The Internet of Things (IoT) is one of the most progressive ideas in the modern era.’ A lot of sentences are poorly constructed. E.g., ‘Bangladesh is an agro-based country and its economy largely depends on it’ – what is ‘it’? ‘is primarily dependent on this agricultural sector’ – what agricultural sector?

Response:

“As per reviewer suggestion we have tried to rewrite this type of poorly constructed sentences.”

Comment: ‘It is said that the land of Bangladesh is fertile and it seems that the farmers are cultivating gold’ – this is not scientific: ‘it is said’, ‘are cultivating gold’. ‘to IoT technology’ – any acronym must be first written in full.

Response:

“As per the reviewer's suggestion we have changed this sentence. The full form of IoT has been written many times in Abstract, so we are not writing its full form everywhere.”

Comment: Sentences are full of word repetitions. E.g., ‘in the agriculture sector has been carried out to develop agricultural benefits and smart agriculture infrastructure’. ‘The application of IoT technology in agriculture has brought great revolutionary changes in the agricultural environment by focusing on multiple challenges and examining different complexities’ – zero info.

Response:

“The first phrase discusses the use of technology to enhance agricultural infrastructure. On the other hand, the second statement clearly discusses the revolutionary IoT technology. And we believe that the meanings of these statements vary on their own.”

Comment: ‘We now hope that with the advancement of this technology, using IoT technology’, ‘Advanced technology is not always, but most of the time benefits people’ – poorly constructed.

Response:

“We have added one more sentence with it to understand why we actually wrote the line, “Advanced technology is not always, but most of the time benefits people”. On the other hand this statement, in our opinion, is appropriate and goes well with the introduction.”

Comment: A lot of ideas are not substantiated. E.g., ‘the IoT has been able to play a major role in everyday life by increasing our perceptions and skills to change the environment around us. IoT is applied in both diagnostics and control, especially in the agro-industrial areas and environmental sectors. It also helps the farmer to provide information about the source and characteristics of the grain or product.’

Response:

“These sentences, in our opinion, are appropriate and reliable. All of the technology we use today has an IoT component. The Internet of Things touches everything, from checking the weather on your phone to using fingerprint sensors, lights, and remotes to operate fans, switches, and motors to pump your water!”

Comment: ‘Later, by analyzing the data in the cloud, various types of forecasts are made available to the farmers’ – ‘later’ when?

Response:

“We have changed the sentence. While changing the sentence, the meaning remained unchanged.”

Comment: ‘how the combination of machine learning, big data, and IoT networks has had a profound effect on farms and agriculture in the Old Testament’ – just checked the paper, it does not refer to the Old Testament. It is strange to relate verifiable data to the Old Testament anyway.

Response:

“We have changed the review of ours thanks to reviewer.”

Comment: The in-text citation system is unusual and not uniform. A lot of cited sources are not developed. ‘They created a control system based on data management and node sensors’ – they who? ‘Their method was put into place to regulate’ – their who?

Response:

“They mean article 8 which is mentioned in the previous sentence. They mean article 9 which is mentioned in the previous sentence. We can't constantly refer to these authors by their names in every phrase in the same way. It can ruin a paper's aesthetic appeal. Each linked work that we have reviewed has been presented in an effort to be unique.”

Comment: ‘A survey of the literature that was centered on studies and analyses of the application of IoT in modern farming’ – incomplete sentence.

Response:

“Don't make a decision based solely on the reference sentence, please. The sentences that follow after all contain the same meaning.”

Comment: ‘Their research and analysis showed how China’ – their who?

Response:

“They mean article 11 which is mentioned in the 2nd next sentence. We can't constantly refer to these authors by their names in every phrase in the same way. It can ruin a paper's aesthetic appeal. Each linked work that we have reviewed has been presented in an effort to be unique.”

Comment: The manuscript is full of such instances. ‘Thebasy overviewed’ – Thebasy?

Response:

“Since no one questioned this word, it went unnoticed. Actually, the error was in the typing. We appreciate the review.”

Comment: The references are not in order and some of them seem to be missing in the text.

  1. ‘In Ref. [17] article authors presented’, ‘In Ref. [32] article authors discusses’ (‘discuss’ anyway) – very unusual.
  2. ‘A literature review on the role of Internet of Things technologies in agriculture that explored the varied effects of IoT in agriculture’ – poorly constructed.
  3. Figures and tables should be improved, unified as style, and thoroughly explained. More development and depth of the methodology and analysis are needed. The reference list is not properly edited.

Response:

 “As suggested by the reviewer, we made an effort to sort the reference numbers. The design and specifics of the figures and tables have been improved.”

Comment: The relationship between cloud-based IoT data analytics and digital twins in smart farming as regards smart crop cultivation systems using automated agriculture monitoring environment  has not been covered, and thus such sources can be cited:

  1. Nica, E.; Popescu, G.H.; Poliak, M.; Kliestik, T.; Sabie, O.-M. Digital Twin Simulation Tools, Spatial Cognition Algorithms, and Multi-Sensor Fusion Technology in Sustainable Urban Governance Networks. Mathematics 2023, 11, 1981. https://doi.org/10.3390/math11091981

  2. Pop, R.A., Dabija, D.C., Pelau, C., Dinu, V. 2022. Usage Intentions, Attitudes, and Behaviours towards Energy-Efficient Applications during the COVID-19 Pandemic. Journal of Business Economics and Management, 23(3), pp.668-689. https://doi.org/10.3846/jbem/2022/16959

  3. Andrei, J. V., Popescu, G. H., Nica, E., & Chivu, L. (2020). The impact of agricultural performance on foreign trade concentration and competitiveness: empirical evidence from Romanian agriculture. Journal of Business Economics and Management, 21(2), 317-343. https://doi.org/10.3846/jbem.2020.11988

Response:

“In our study, we cited and briefly discussed the first reference. We thought it should be noted because it is somewhat similar to our scope. And we apologize for the others; we did not cite them because there was no fit with our scope.”

Comments on the Quality of English Language

A lot of sentences are poorly constructed. E.g., ‘Bangladesh is an agro-based country and its economy largely depends on it’ – what is ‘it’? ‘is primarily dependent on this agricultural sector’ – what agricultural sector? ‘We now hope that with the advancement of this technology, using IoT technology’, ‘Advanced technology is not always, but most of the time benefits people’ – poorly constructed. ‘They created a control system based on data management and node sensors’ – they who? ‘Their method was put into place to regulate’ – their who? ‘A survey of the literature that was centered on studies and analyses of the application of IoT in modern farming’ – incomplete sentence. ‘Their research and analysis showed on how China’ – their who? The manuscript is full of such instances. ‘Thebasy overviewed’ – Thebasy? ‘A literature review on the role of Internet of Things technologies in agriculture that explored the varied effects of IoT in agriculture’ – poorly constructed.

Response:

“We have tried to improve the quality of the English language according to the reviewer's words.”

Other Reviewer Suggestions and our Response:

Reviewer 01 comments and our feedback:

Smart crop cultivation system using automated agriculture monitoring environment in the context of Bangladesh Agriculture is presented in this work and following are the observations and comments from this work that is to be addressed,

  1. This work seems to be novel from the country-specific but such works have been done across the globe for the past many years. So novelty perspective should be strongly illustrated.

Response:

“The characteristics developed here are country-specific innovations, and many other regions of the world already use them. Nobody has yet to come across a scientific article that has all these traits. Additionally, we have talked about the analysis of datasets that are country-specific. The primary contributions listed in the introduction are also covered in detail.”

  1. Why is clustering used in this work? Is this about the unlabeled data collected?

Response:

“We have used cluster in this work because initially when we collected the data and collected a dataset, the data was very unlabeled and had similar characteristics data. As a result we do clustering for preprocessing. This process of clustering is described in two broad parts. The first part is covered in Sub-Section 3.2.3 (II) and the second part is mentioned first in Section 4.4 Data Curation and Results.”

  1. Details of the attributes measured using sensors should be separately stated.

Response:

“In Sub Section 3.2. System Implementation, 3.2.1. Crops data generation, real-time forecasting, and empirical setup section, the properties measured by the sensors are covered in depth.”

  1. How forecasting is done? What will be forecasted?

Response:

“In order to determine which crops can be planted and which crops can be harvested in the continuing monthly cycle, our system architecture gathers real-time data from sensors and analyzes it alongside historical data. Additionally, by integrating APIs, the area where the farmer's cropland is located can predict the weather in real time for the next five days.”

  1. Forecasting algorithm is novel?

Response:

“Undoubtedly it is novel.”

  1. How it is different from other works?

Response:

“Numerous nations currently and in the future carry out this type of work. There is no confirmation that a farmer will have access to all the characteristics in one application, even though more or less similar features have been observed in any study conducted thus far. However, our system structure is flexible and packed with features. It must be merged with the country's statistical historical data in order to be used in the agriculture sector of any country. However, South East Asian nations may easily implement our system framework.”

  1. There are no comparison results with any state of the art of techniques.

Response:

“We don't think that it is necessary to provide separate comparative results for the techniques others used and what we used for data analysis and the programs utilized as input to the devices. It's also unclear whether or not it will fit with our paper. Also, if we were considering this paper as a case study or a review paper, that table or result might be required.”

  1. There are plenty of works done with respect to IoT + cloud + forecasting that can be compared and the result of a web application or a mobile application may not be a significant outcome of the research.

Response:

“We already did it in Table 01.”

  1. This work needs to be significantly improved from the research perspective.

Response:

“We have made a lot of effort to improve our job. In the revolutionary IoT sector, fresh research will be conducted continuously to advance technology. The touch of better technology will be accessible when the price of IoT industry devices increases. However, we have made an effort to provide as many functions as we can in a web or mobile application while keeping below a realistic cost, keeping farmers in mind. We wrote this research paper to address issues such as how to automate farming, forecast crops and weather, monitor farm equipment in real time, and show how a farmer can use our framework to develop opportunities for producing eco-friendly crops without the assistance of an agricultural officer. Many things can be integrated if desired. However, we have tried to squeeze all the IoT devices available in this region and as many resources as we can explore. There will be more work on this in the future. But as of now, no one can see such a numerical feature. We strongly hope that people who work on it in the future will find our paper to be among the most pertinent ones.”

Reviewer 03 comments and our revisions:

Manuscript

Title: „Smart crop cultivation system using automated agriculture monitoring environment in the context of Bangladesh Agriculture”

Authors: Md. Bayazid Rahman , Joy Dhon Chakma, Dr. Abdul Momin , Dr. Shahidul Islam , Dr. Md Ashraf Uddin , Dr. Md Aminul Islam

Dear Authors

I revised the manuscript: "Smart crop cultivation system using automated agriculture monitoring environment in the context of Bangladesh Agriculture” submitted to the „Sensors” Journal. The paper is very interesting. However, I have some concerns, which need to be addressed.

Line 2-4. Article topic

The theme of the article is concise and accurately reflects the content of the article.

The structure of the article, divided into chapters and subchapters, is clear and logical.

It is noticeable that there are a lack of keywords in the topic of the article, especially: ”prediction, data mining, internet of things etc”.

This means the broader context of the research findings. In my opinion, however, the content of the article is a form of case study. Please consider converting the article topic to better represent the content of the article.

Response:

“As per the reviewer's suggestion, the issues that we have worked on in the research have been changed and updated in the keyword section. Lines 24-25.”

Abstract

Line 13-23. The content of the abstract should indicate the measurable effect of the research in terms of giving a numerical value and indicating the most important conclusions generalised even within the implementation of the case study. Please take this into account.

The abstract is a self-contained part of the article which requires repetition of any explanatory notes.

The authors suggest the goal and scope of the work, but the main goal of the work is split between individual specific tasks, which are shown implicitly as sub-results. The descriptions of the results are very abbreviated.

Line 20-21. „….several experiments are adopted to check the….” The expression is imprecise and should have no place in the abstract.

It is difficult to identify the leading result and the leading conclusion of the research. Please take this into account.

Response:

“We follow the reviewer's suggestion „….several experiments are adopted to check the….” this sentence is omitted. Also, we have tried to highlight our work as much as possible in the abstract with leading results and conclusions. For which we try to make changes according to the suggestions of the reviewer in the abstract.”

Keywords. Line 24-25. The authors used the full spectrum of matching wording. Keywords contain word clusters and are too literal, but represents the spectrum of information well. The order of keywords should follow the concept of „from generalities to specifics”. Keywords may also reflect the order in which the research issues are addressed.  Please consider changing the order of keywords.

Please try to arrange your keywords according to a coherent concept and dispense with near-meaningful expressions and duplicate messages.

Response:

“As per the reviewer suggestion, the issues that we have worked on in the research have been changed and updated in the keyword section. Lines 24-25.”

  1. Introduction

The state of the knowledge presented is relevant to the goal and scope of the research. The presentation of the state of knowledge in the chapter is quite general and its form is quite brief. The authors have used only four literature sources to outline the state of research in the spectrum discussed. Please consider elaborating the content provided in more detail in the context of an attractive and internationally recognized research topic.

Response:

“As per reviewer's suggestion we have added more references in “Introduction” Chapter.”

Line 65-69. The included information indicates research results related to a case study analysis (case study), with limited possibilities of generalization. This is a conclusion and should conclude the analysis of the research results instead making a form of hypothesis.

Response:

“As per reviewer's suggestion we have added few more details.”

Line 70 – 76. A formal presentation of the content division of the article is not necessary. Please take this into consideration.

Response:

“Based on the other reference articles format style, we decided to mention the arrangement of article for clarity of the readers.”

  1. Related Work

The lack of an indication of the goal and scope of the research in Chapter 1, even in a descriptive and general way, makes it difficult for the reader to understand the intention of the content introduced in Chapter 2. The reader has the impression of knowing rather general and difficult to relate thematic content. A further analysis of the state of the knowledge is continued in the next chapter. This division is not necessary.

The order of citation of the literature sources coinciding with the ordinal numbers is disrupted because literature source with ordinal number [5] only appear in line 233. Please discuss this with the journal editors or correct the problem.

Response:

“As per reviewer suggestion we shuffle and corrected the order of the sources number according to sensor journal format.”

The order of citation of the literature sources coinciding with the ordinal numbers is disrupted because literature source with ordinal number [15] only appear in line 414. Please discuss this with the journal editors or correct the problem.

Response:

“As per reviewer suggestion we shuffle and corrected the order of the sources number according to sensor journal format.”

The order of citation of the literature sources coinciding with the ordinal numbers is disrupted because literature items with ordinal numbers [24] to [31] only appear in lines: Line 403 [24], Line 368 [25], Line 233 [31] . Please discuss this with the journal editors or correct the problem.

Response: “As per reviewer suggestion we shuffle and corrected the order of the sources number according to sensor journal format.”

The literature source number [26] was not self-quoted in the text of the article in the correct order. Please correct this. (only then in Line 384)

Response: “As per reviewer suggestion we corrected to the standard MDPI Sensors format.”

The literature source number [27] was not self-quoted in the text of the article in the correct order. Please correct this. (only then in Line 384)

Response: “As per reviewer suggestion we corrected to the standard MDPI Sensors format.”

The literature source number [28] was not self-quoted in the text of the article in the correct order. Please correct this. (only then in Line 384)

Response: “As per reviewer suggestion we corrected to the standard MDPI Sensors format.”

The literature source number [29] was not self-quoted in the text of the article in the correct order. Please correct this. (only then in Line 451)

Response: “As per reviewer suggestion we corrected to the standard MDPI Sensors format.”

The literature source number [30] was not self-quoted in the text of the article in the correct order. Please correct this. (only then in Line 451)

Response: “As per reviewer suggestion we corrected to the standard MDPI Sensors format.”

Line 80. „….Andreas et al. [7] provided a literature review demonstrating….” We do not refer to a 'literature review' but to 'research results'. Please take this into account.

Response: “As per reviewer suggestion we have changed it from “Literature review” to “study of research”.” 

Line 97. „….RFID…” Please explain the introduced abbreviations of terms, possibly as soon as they are introduced in the content of the article. Even popular acronym names need to be explained in scientific articles.  A single explanation is sufficient in the execution of the explanation.

Response:

“As per reviewer suggestion we have this abbreviation to full form.”

Line 100. „….As surveyed in [13]….”This formulation is too simplistic. A ordinal number cannot be implicitly considered as a source of knowledge. The sentence should be completed with the words: scientific article.

Response:

“As per reviewer suggestion we have changed it into “scientific article”.”

Line 103 – 122. Research results or scopes of research of analyzed scientific articles should be cited in the context of convergence with the own research task. In this sense, the fragment of chapter two and many other parts are too general and weakly linked to the goal and scope of your own research.  Please correct this.

Response:

“We remove a citation from our paper. We believe that the referenced work is considerably distinct from our scope and goals, as suggested by the reviewer.

The removed paper:

RL, Raghavi, and A. Umamageswari. "Modern Irrigation based on Web Weather Forecast." (2018).”

Line 146. „….The AgriTrust….” Please explain the introduced abbreviations of terms, possibly as soon as they are introduced in the content of the article. Even popular acronym names need to be explained in scientific articles.  A single explanation is sufficient in the execution of the explanation.

Response:

“As per reviewer suggestion we added the explanation of abbreviation “AgriTrust”.”

Line 160. Table 1. Comparative features and criteria appear too late for understanding the intention and tasks of the presented scientific article. Please introduce the criteria and scopes of your own research earlier in the content of the article.

Lack of reference to 'Table 1' in the content of chapter two. Please correct this if Table 1 should stay in the content of Chapter Two.

Response:

“Thank you for noticing it. We move the table in a suitable chapter where it actually belong. We move it to sub section 3.2. (System Implementation).”

  1. Our Empirical Approach

Line 175-178. „…The purpose of this work is to send data from the farm to the web application or smart phone through the data management controller named NodeMCU. The design and implementation overview of this system is divided into 3 components namely hardware, web/mobile application, and cloud database, as shown in Figure 2…. ”

There is a noticeable dissonance between: the theoretical, informative preparation of Chapters 1 and 2, the research methods, the verification content of Chapter 4, and the notation from lines 175 to 178 of Chapter 3. The highlighted section of the chapter contains a mental shortcut that requires two actions: a detailed explanation of the goal and scope of the research and the creation of research methods and results logically connected to the stated goal and scope of the research. It should also be noted about the order of presentation of the research results, which depends on the logical sequence of the appearance of the sub-tasks and the support of the research methods and research stands. Please take this into account in the processes of organising the content of the scientific article.

Response:

“According to the reviewer's point of view, we have tried to cover the inconsistency of the theoretical and informative preparation in the first paragraph of Chapter 3.1. We have added some new information to this paragraph to resolve this inconsistency.”

Line 176. „….NodeMCU….” Please explain the introduced abbreviations of terms, possibly as soon as they are introduced in the content of the article. Even popular acronym names need to be explained in scientific articles.  A single explanation is sufficient in the execution of the explanation.

Response:

“As per reviewer suggestion we added the explanation of abbreviation “NodeMCU” and “ESP8266”.”

Line 183, 220, 249, 488, . „….DHT 11….” Please explain the introduced abbreviations of terms, possibly as soon as they are introduced in the content of the article. Even popular acronym names need to be explained in scientific articles.  A single explanation is sufficient in the execution of the explanation.

Response:

“We have added the explanation of “what DHT 11 device is.” in a single explanation in our paper. We have only added the explanation to the first abbreviation. Abbreviations are left in the rest only. Since the explanation has been done in the very first one, we think that it does not need an explanation in the latter ones.”

Line 183. „….a node MCU…” Please explain the introduced abbreviations of terms, possibly as soon as they are introduced in the content of the article. Even popular acronym names need to be explained in scientific articles.  A single explanation is sufficient in the execution of the explanation.

Response:

“We have added the explanation of NodeMCU in chapter 3.1 as per suggestion of reviewer. Since the explanation has been done in the very first one, we think that it does not need an explanation in the latter ones.”

Line 192. „….MIT…..” Please explain the introduced abbreviations of terms, possibly as soon as they are introduced in the content of the article. Even popular acronym names need to be explained in scientific articles.  A single explanation is sufficient in the execution of the explanation.

Response:

“We have added the explanation of “what MIT app inventor is.””

Line 196. „….API….” Please explain the introduced abbreviations of terms, possibly as soon as they are introduced in the content of the article. Even popular acronym names need to be explained in scientific articles.  A single explanation is sufficient in the execution of the explanation.

Response:

“We have added the full form of “API.””

Line 204. „…Wi-Fi network…” Please explain the introduced abbreviations of terms, possibly as soon as they are introduced in the content of the article. Even popular acronym names need to be explained in scientific articles.  A single explanation is sufficient in the execution of the explanation. For example: (wireless fidelity)

Response:

“We have added the full form of “Wi-Fi””

Line 206. „…IPv6….” Please explain the introduced abbreviations of terms, possibly as soon as they are introduced in the content of the article. Even popular acronym names need to be explained in scientific articles.  A single explanation is sufficient in the execution of the explanation.

Response:

“We have added the full form of “IPv6””

Line 201 - 215: "...(i), (ii), (iii), (iv), (v), (vi), (vii)...". Please indicate the proposed layer identifiers as designations in Figure 5. The notation is incomprehensible. Please correct it.

Response:

“We updated figure 5 as per reviewer suggestion.”

Line 221. „….ESP-8266 module….” Please explain the introduced abbreviations of terms, possibly as soon as they are introduced in the content of the article. Even popular acronym names need to be explained in scientific articles.  A single explanation is sufficient in the execution of the explanation.

Response:

“We already explained it before in chapter 3.1 as per reviewer suggestion.”

Line 237. „…ESP8266 Node MCU…” The notation of the same acronym designations will change in the body of the article. Please standardise the notation of names and acronyms.

Response:

“We already explained it before in chapter 3.1 as per reviewer suggestion.”

Line 369. „…(i)….” Why do the authors not number the mathematical formulae with numerical designations? There is an apparent conflict with the description markings of Figure 5.

Response:

“We have changed the mathematical formulae numerical to number designation.”

Line 385. „….(ii)…..” Why do the authors not number the mathematical formulae with numerical designations? There is an apparent conflict with the description markings of Figure 5.

Response:

“We have changed the mathematical formulae numerical to number designation.”

Line 391. Figure 7. The content of the textual information in the figure is unreadable. Please correct this.

Response:

“We have changed the figure and try to make textual information in the figure is readable.”

Line 394. Figure 8. There is a lack of reference in the body of the article to Figure 8, in the assigned chapter to Figure 8. Please complete this.

The content of the textual information in the figure is unreadable. Please correct this.

Response:

“We have changed the figure and try to make textual information in the figure is readable. And also add a reference for this that we forgot to add in this figure 8 (5 no.). Thank you for that.”

  1. Experimental Setup and result discussion

The assumptions presented for the study material and research methods, when confronted with the parameter assumptions in Table 1, are difficult to harmonise and reconcile. There is a lack of clear separation between the scope of the research and the coinciding results.

Response:

“Our study and research work is heavily defends on the application layer because of IoT works and physical layer. And both comes together perfectly with the associated layers. And from the associated layer one of the important layer to get accuracy and also show the correct result to farmers is the configuration layer where we did data analysis, data acquisition and data management configuration. In table 1 the features are came from application layer. Actually when we think about farmers is that actually how much feature we are giving them. So that we compare other scientific paper application layers feature with our application layer feature. As per reviewer suggestion we moved this table 1 from chapter 2 and placed it in chapter 3.2.”

Line 429. „…SOSE….” Sum of the squared errors, SSE, is defined as follows: SSE = N ∑ ei2 =∑ (xi − ˆxi)2. Is this the same notation as the SOSE parameter? There is no explicit definition of the SOSE parameter in the form of a mathematical model.

Response:

“As per reviewer suggestion we explained the SOSE parameter with a mathematical model.”

Line 429. „…. incorrectly clustered instances (ICI)….” There is a lack of an explicit definition of the ICI parameter in the form of a mathematical model.

Response:

“As per reviewer suggestion we explained the ICI parameter with a mathematical model.”

Line 434. „…. ISI ….” Please explain the introduced abbreviations of terms, possibly as soon as they are introduced in the content of the article. Even popular acronym names need to be explained in scientific articles.  A single explanation is sufficient in the execution of the explanation.

Response:

“We already have the explanation of ICI in cluster section as per reviewer suggestion.”

Line 442. Figure 9. Lack of markings (a) and (b) on the individual graphics of Figure 9.

Response:

“Thanks to reviewer we updated the figure 9.”

Line 457. „….Table 03…” This is a multiplication of the synonyms of Table 3. Please homogenise the notation. Please do not create alternative designations for the same element.

Response:

“Thanks to reviewer we updated the designations of table 3 and other notations.”

Line 463. Table 3. „…TP Rate, FP Rate, Precision, Recall, F-Measure, ROC Area,….” Please explain the acronyms introduced in the table independently of the content of the chapter.

Response:

“We add each of the acronyms full form in Table 3.”

  1. Discussion & Future Work

Line 512, 514. „…MQTT….” Please explain the introduced abbreviations of terms, possibly as soon as they are introduced in the content of the article. Even popular acronym names need to be explained in scientific articles.  A single explanation is sufficient in the execution of the explanation.

Response:

“We added the acronym MQTT full form.”

Line 492 - 509 The content fits more with Chapters 1 and 2 and should supply the goal and scope of the work. Please take this into account.

Response:

“As per reviewer suggestion we move the first 4 to 5 sentence in chapter 1 introduction. And we added there a valuable discussion about our work.”

Line 510-521. „….Our main challenge in this work is to collect data on crop yield, planting and season nformation in Bangladesh….”This formulation reminds the scope of the study but is not articulated earlier. It is useful to organise the content of the article and create a logical sequence of statements, results and conclusions. Please take this into account.

Response:

“We have not varied much in our challenge. We particularly believe that our challenges are valid and that the counterarguments to the challenges are valid.”

The content presented in Chapter 5 does not constitute a valuable discussion of the results because, as in Chapter 4, the range of comparative research findings cited from the analysis of the state of knowledge is insufficient even for a so-called case study. Please complete and expand the discussion of the results.

Response:

“We regret that Chapter 5 lacks constructive discussion. We have tried to make the whole discussion part different according to the reviewer's suggestion and have done many constructive discussions. Hope the reviewer will enjoy reading it.”

  1. Conclusion

Line 523 - 530. The content of the chapter matches the opening of the discussion about results rather than the conclusion.

Line 531 – 539. The second part of Chapter 6 better represents the concept of the conclusion.

The presentation of results in this second part of the chapter is adequate to present the conclusions. Please take this into account.

Response:

“As per reviewer comments, we removed the contents that matches with other chapters content. And also we kept only second part as per reviewer suggestion.”

References

Line 540-624. Please check for literature sources available online. Please verify whether other forms of availability and other forms of publisher addresses for online publication sources can be indicated.

Response:

“We investigated, and their availability is consistent.”

Reviewer 3 Report

Manuscript

Title:Smart crop cultivation system using automated agriculture monitoring environment in the context of Bangladesh Agriculture”

Authors: Md. Bayazid Rahman , Joy Dhon Chakma, Dr. Abdul Momin , Dr. Shahidul Islam , Dr. Md Ashraf Uddin , Dr. Md Aminul Islam

Dear Authors

I revised the manuscript: "Smart crop cultivation system using automated agriculture monitoring environment in the context of Bangladesh Agriculture” submitted to the „Sensors” Journal. The paper is very interesting. However, I have some concerns, which need to be addressed.

Line 2-4. Article topic

The theme of the article is concise and accurately reflects the content of the article.

The structure of the article, divided into chapters and subchapters, is clear and logical.

It is noticeable that there are a lack of keywords in the topic of the article, especially: ”prediction, data mining, internet of things etc”.

This means the broader context of the research findings. In my opinion, however, the content of the article is a form of case study. Please consider converting the article topic to better represent the content of the article.

Abstract

Line 13-23. The content of the abstract should indicate the measurable effect of the research in terms of giving a numerical value and indicating the most important conclusions generalised even within the implementation of the case study. Please take this into account.

The abstract is a self-contained part of the article which requires repetition of any explanatory notes.

The authors suggest the goal and scope of the work, but the main goal of the work is split between individual specific tasks, which are shown implicitly as sub-results. The descriptions of the results are very abbreviated.

Line 20-21. „….several experiments are adopted to check the….” The expression is imprecise and should have no place in the abstract.

It is difficult to identify the leading result and the leading conclusion of the research. Please take this into account.

Keywords. Line 24-25. The authors used the full spectrum of matching wording. Keywords contain word clusters and are too literal, but represents the spectrum of information well. The order of keywords should follow the concept of „from generalities to specifics”. Keywords may also reflect the order in which the research issues are addressed.  Please consider changing the order of keywords.

Please try to arrange your keywords according to a coherent concept and dispense with near-meaningful expressions and duplicate messages.

1. Introduction

The state of the knowledge presented is relevant to the goal and scope of the research. The presentation of the state of knowledge in the chapter is quite general and its form is quite brief. The authors have used only four literature sources to outline the state of research in the spectrum discussed. Please consider elaborating the content provided in more detail in the context of an attractive and internationally recognised research topic.

Line 65-69. The included information indicates research results related to a case study analysis (case study), with limited possibilities of generalisation. This is a conclusion and should conclude the analysis of the research results instead making a form of hypothesis.

Line 70 – 76. A formal presentation of the content division of the article is not necessary. Please take this into consideration.

2. Related Work

The lack of an indication of the goal and scope of the research in Chapter 1, even in a descriptive and general way, makes it difficult for the reader to understand the intention of the content introduced in Chapter 2. The reader has the impression of knowing rather general and difficult to relate thematic content. A further analysis of the state of the knowledge is continued in the next chapter. This division is not necessary.

The order of citation of the literature sources coinciding with the ordinal numbers is disrupted because literature source with ordinal number [5] only appear in line 233. Please discuss this with the journal editors or correct the problem.

The order of citation of the literature sources coinciding with the ordinal numbers is disrupted because literature source with ordinal number [15] only appear in line 414. Please discuss this with the journal editors or correct the problem.

The order of citation of the literature sources coinciding with the ordinal numbers is disrupted because literature items with ordinal numbers [24] to [31] only appear in lines: Line 403 [24], Line 368 [25], Line 233 [31] . Please discuss this with the journal editors or correct the problem.

The literature source number [26] was not self-quoted in the text of the article in the correct order. Please correct this. (only then in Line 384)

The literature source number [27] was not self-quoted in the text of the article in the correct order. Please correct this. (only then in Line 384)

The literature source number [28] was not self-quoted in the text of the article in the correct order. Please correct this. (only then in Line 384)

The literature source number [29] was not self-quoted in the text of the article in the correct order. Please correct this. (only then in Line 451)

The literature source number [30] was not self-quoted in the text of the article in the correct order. Please correct this. (only then in Line 451)

Line 80. „….Andreas et al. [7] provided a literature review demonstrating….” We do not refer to a 'literature review' but to 'research results'. Please take this into account.

Line 97. „….RFID…” Please explain the introduced abbreviations of terms, possibly as soon as they are introduced in the content of the article. Even popular acronym names need to be explained in scientific articles.  A single explanation is sufficient in the execution of the explanation.

Line 100. „….As surveyed in [13]….”This formulation is too simplistic. A ordinal number cannot be implicitly considered as a source of knowledge. The sentence should be completed with the words: scientific article.

Line 103 – 122. Research results or scopes of research of analysed scientific articles should be cited in the context of convergence with the own research task. In this sense, the fragment of chapter two and many other parts are too general and weakly linked to the goal and scope of your own research.  Please correct this.

Line 146. „….The AgriTrust….” Please explain the introduced abbreviations of terms, possibly as soon as they are introduced in the content of the article. Even popular acronym names need to be explained in scientific articles.  A single explanation is sufficient in the execution of the explanation.

Line 160. Table 1. Comparative features and criteria appear too late for understanding the intention and tasks of the presented scientific article. Please introduce the criteria and scopes of your own research earlier in the content of the article.

Lack of reference to 'Table 1' in the content of chapter two. Please correct this if Table 1 should stay in the content of Chapter Two.

3. Our Empirical Approach

Line 175-178. „…The purpose of this work is to send data from the farm to the web application or smart phone through the data management controller named NodeMCU. The design and implementation overview of this system is divided into 3 components namely hardware, web/mobile application, and cloud database, as shown in Figure 2….”

There is a noticeable dissonance between: the theoretical, informative preparation of Chapters 1 and 2, the research methods, the verification content of Chapter 4, and the notation from lines 175 to 178 of Chapter 3. The highlighted section of the chapter contains a mental shortcut that requires two actions: a detailed explanation of the goal and scope of the research and the creation of research methods and results logically connected to the stated goal and scope of the research. It should also be noted about the order of presentation of the research results, which depends on the logical sequence of the appearance of the sub-tasks and the support of the research methods and research stands. Please take this into account in the processes of organising the content of the scientific article.

Line 176. „….NodeMCU….” Please explain the introduced abbreviations of terms, possibly as soon as they are introduced in the content of the article. Even popular acronym names need to be explained in scientific articles.  A single explanation is sufficient in the execution of the explanation.

Line 183, 220, 249, 488, . „….DHT 11….” Please explain the introduced abbreviations of terms, possibly as soon as they are introduced in the content of the article. Even popular acronym names need to be explained in scientific articles.  A single explanation is sufficient in the execution of the explanation.

Line 183. „….a node MCU…” Please explain the introduced abbreviations of terms, possibly as soon as they are introduced in the content of the article. Even popular acronym names need to be explained in scientific articles.  A single explanation is sufficient in the execution of the explanation.

Line 192. „….MIT…..” Please explain the introduced abbreviations of terms, possibly as soon as they are introduced in the content of the article. Even popular acronym names need to be explained in scientific articles.  A single explanation is sufficient in the execution of the explanation.

Line 196. „….API….” Please explain the introduced abbreviations of terms, possibly as soon as they are introduced in the content of the article. Even popular acronym names need to be explained in scientific articles.  A single explanation is sufficient in the execution of the explanation.

Line 204. „…Wi-Fi network…” Please explain the introduced abbreviations of terms, possibly as soon as they are introduced in the content of the article. Even popular acronym names need to be explained in scientific articles.  A single explanation is sufficient in the execution of the explanation. For example: (wireless fidelity)

Line 206. „…IPv6….” Please explain the introduced abbreviations of terms, possibly as soon as they are introduced in the content of the article. Even popular acronym names need to be explained in scientific articles.  A single explanation is sufficient in the execution of the explanation.

Line 201 - 215: "...(i), (ii), (iii), (iv), (v), (vi), (vii)...". Please indicate the proposed layer identifiers as designations in Figure 5. The notation is incomprehensible. Please correct it.

Line 221. „….ESP-8266 module….” Please explain the introduced abbreviations of terms, possibly as soon as they are introduced in the content of the article. Even popular acronym names need to be explained in scientific articles.  A single explanation is sufficient in the execution of the explanation.

Line 237. „…ESP8266 Node MCU…” The notation of the same acronym designations will change in the body of the article. Please standardise the notation of names and acronyms.

Line 369. „…(i)….” Why do the authors not number the mathematical formulae with numerical designations? There is an apparent conflict with the description markings of Figure 5.

Line 385. „….(ii)…..” Why do the authors not number the mathematical formulae with numerical designations? There is an apparent conflict with the description markings of Figure 5.

Line 391. Figure 7. The content of the textual information in the figure is unreadable. Please correct this.

Line 394. Figure 8. There is a lack of reference in the body of the article to Figure 8, in the assigned chapter to Figure 8. Please complete this.

The content of the textual information in the figure is unreadable. Please correct this.

4. Experimental Setup and result discussion

The assumptions presented for the study material and research methods, when confronted with the parameter assumptions in Table 1, are difficult to harmonise and reconcile. There is a lack of clear separation between the scope of the research and the coinciding results.

Line 429. „…SOSE….” Sum of the squared errors, SSE, is defined as follows: SSE = N ∑ ei2 =∑ (xi − ˆxi)2. Is this the same notation as the SOSE parameter? There is no explicit definition of the SOSE parameter in the form of a mathematical model.

Line 429. „…. incorrectly clustered instances (ICI)….” There is a lack of an explicit definition of the ICI parameter in the form of a mathematical model.

Line 434. „…. ISI ….” Please explain the introduced abbreviations of terms, possibly as soon as they are introduced in the content of the article. Even popular acronym names need to be explained in scientific articles.  A single explanation is sufficient in the execution of the explanation.

Line 442. Figure 9. Lack of markings (a) and (b) on the individual graphics of Figure 9.

Line 457. „….Table 03…” This is a multiplication of the synonyms of Table 3. Please homogenise the notation. Please do not create alternative designations for the same element.

Line 463. Table 3. „…TP Rate, FP Rate, Precision, Recall, F-Measure, ROC Area,….” Please explain the acronyms introduced in the table independently of the content of the chapter.

5. Discussion & Future Work

Line 512, 514. „…MQTT….” Please explain the introduced abbreviations of terms, possibly as soon as they are introduced in the content of the article. Even popular acronym names need to be explained in scientific articles.  A single explanation is sufficient in the execution of the explanation.

Line 492 - 509 The content fits more with Chapters 1 and 2 and should supply the goal and scope of the work. Please take this into account.

Line 510-521. „….Our main challenge in this work is to collect data on crop yield, planting and season nformation in Bangladesh….”This formulation reminds the scope of the study but is not articulated earlier. It is useful to organise the content of the article and create a logical sequence of statements, results and conclusions. Please take this into account.

The content presented in Chapter 5 does not constitute a valuable discussion of the results because, as in Chapter 4, the range of comparative research findings cited from the analysis of the state of knowledge is insufficient even for a so-called case study. Please complete and expand the discussion of the results.

6. Conclusion

Line 523 - 530. The content of the chapter matches the opening of the discussion about results rather than the conclusion.

Line 531 – 539. The second part of Chapter 6 better represents the concept of the conclusion.

The presentation of results in this second part of the chapter is adequate to present the conclusions. Please take this into account.

References

Line 540-624. Please check for literature sources available online. Please verify whether other forms of availability and other forms of publisher addresses for online publication sources can be indicated.

Author Response

The reviewer comments and our revisions:

Manuscript

Title: „Smart crop cultivation system using automated agriculture monitoring environment in the context of Bangladesh Agriculture”

Authors: Md. Bayazid Rahman , Joy Dhon Chakma, Dr. Abdul Momin , Dr. Shahidul Islam , Dr. Md Ashraf Uddin , Dr. Md Aminul Islam

Dear Authors

I revised the manuscript: "Smart crop cultivation system using automated agriculture monitoring environment in the context of Bangladesh Agriculture” submitted to the „Sensors” Journal. The paper is very interesting. However, I have some concerns, which need to be addressed.

Line 2-4. Article topic

The theme of the article is concise and accurately reflects the content of the article.

The structure of the article, divided into chapters and subchapters, is clear and logical.

It is noticeable that there are a lack of keywords in the topic of the article, especially: ”prediction, data mining, internet of things etc”.

This means the broader context of the research findings. In my opinion, however, the content of the article is a form of case study. Please consider converting the article topic to better represent the content of the article.

Response:

“As per the reviewer's suggestion, the issues that we have worked on in the research have been changed and updated in the keyword section. Lines 24-25.”

Abstract

Line 13-23. The content of the abstract should indicate the measurable effect of the research in terms of giving a numerical value and indicating the most important conclusions generalised even within the implementation of the case study. Please take this into account.

The abstract is a self-contained part of the article which requires repetition of any explanatory notes.

The authors suggest the goal and scope of the work, but the main goal of the work is split between individual specific tasks, which are shown implicitly as sub-results. The descriptions of the results are very abbreviated.

Line 20-21. „….several experiments are adopted to check the….” The expression is imprecise and should have no place in the abstract.

It is difficult to identify the leading result and the leading conclusion of the research. Please take this into account.

Response:

“We follow the reviewer's suggestion „….several experiments are adopted to check the….” this sentence is omitted. Also, we have tried to highlight our work as much as possible in the abstract with leading results and conclusions. For which we try to make changes according to the suggestions of the reviewer in the abstract.”

Keywords. Line 24-25. The authors used the full spectrum of matching wording. Keywords contain word clusters and are too literal, but represents the spectrum of information well. The order of keywords should follow the concept of „from generalities to specifics”. Keywords may also reflect the order in which the research issues are addressed.  Please consider changing the order of keywords.

Please try to arrange your keywords according to a coherent concept and dispense with near-meaningful expressions and duplicate messages.

Response:

“As per the reviewer suggestion, the issues that we have worked on in the research have been changed and updated in the keyword section. Lines 24-25.”

  1. Introduction

The state of the knowledge presented is relevant to the goal and scope of the research. The presentation of the state of knowledge in the chapter is quite general and its form is quite brief. The authors have used only four literature sources to outline the state of research in the spectrum discussed. Please consider elaborating the content provided in more detail in the context of an attractive and internationally recognized research topic.

Response:

“As per reviewer's suggestion we have added more references in “Introduction” Chapter.”

Line 65-69. The included information indicates research results related to a case study analysis (case study), with limited possibilities of generalization. This is a conclusion and should conclude the analysis of the research results instead making a form of hypothesis.

Response:

“As per reviewer's suggestion we have added few more details.”

Line 70 – 76. A formal presentation of the content division of the article is not necessary. Please take this into consideration.

Response:

“Based on the other reference articles format style, we decided to mention the arrangement of article for clarity of the readers.”

  1. Related Work

The lack of an indication of the goal and scope of the research in Chapter 1, even in a descriptive and general way, makes it difficult for the reader to understand the intention of the content introduced in Chapter 2. The reader has the impression of knowing rather general and difficult to relate thematic content. A further analysis of the state of the knowledge is continued in the next chapter. This division is not necessary.

The order of citation of the literature sources coinciding with the ordinal numbers is disrupted because literature source with ordinal number [5] only appear in line 233. Please discuss this with the journal editors or correct the problem.

Response:

“As per reviewer suggestion we shuffle and corrected the order of the sources number according to sensor journal format.”

The order of citation of the literature sources coinciding with the ordinal numbers is disrupted because literature source with ordinal number [15] only appear in line 414. Please discuss this with the journal editors or correct the problem.

Response:

“As per reviewer suggestion we shuffle and corrected the order of the sources number according to sensor journal format.”

The order of citation of the literature sources coinciding with the ordinal numbers is disrupted because literature items with ordinal numbers [24] to [31] only appear in lines: Line 403 [24], Line 368 [25], Line 233 [31] . Please discuss this with the journal editors or correct the problem.

Response: “As per reviewer suggestion we shuffle and corrected the order of the sources number according to sensor journal format.”

The literature source number [26] was not self-quoted in the text of the article in the correct order. Please correct this. (only then in Line 384)

Response: “As per reviewer suggestion we corrected to the standard MDPI Sensors format.”

The literature source number [27] was not self-quoted in the text of the article in the correct order. Please correct this. (only then in Line 384)

Response: “As per reviewer suggestion we corrected to the standard MDPI Sensors format.”

The literature source number [28] was not self-quoted in the text of the article in the correct order. Please correct this. (only then in Line 384)

Response: “As per reviewer suggestion we corrected to the standard MDPI Sensors format.”

The literature source number [29] was not self-quoted in the text of the article in the correct order. Please correct this. (only then in Line 451)

Response: “As per reviewer suggestion we corrected to the standard MDPI Sensors format.”

The literature source number [30] was not self-quoted in the text of the article in the correct order. Please correct this. (only then in Line 451)

Response: “As per reviewer suggestion we corrected to the standard MDPI Sensors format.”

Line 80. „….Andreas et al. [7] provided a literature review demonstrating….” We do not refer to a 'literature review' but to 'research results'. Please take this into account.

Response: “As per reviewer suggestion we have changed it from “Literature review” to “study of research”.” 

Line 97. „….RFID…” Please explain the introduced abbreviations of terms, possibly as soon as they are introduced in the content of the article. Even popular acronym names need to be explained in scientific articles.  A single explanation is sufficient in the execution of the explanation.

Response:

“As per reviewer suggestion we have this abbreviation to full form.”

Line 100. „….As surveyed in [13]….”This formulation is too simplistic. A ordinal number cannot be implicitly considered as a source of knowledge. The sentence should be completed with the words: scientific article.

Response:

“As per reviewer suggestion we have changed it into “scientific article”.”

Line 103 – 122. Research results or scopes of research of analyzed scientific articles should be cited in the context of convergence with the own research task. In this sense, the fragment of chapter two and many other parts are too general and weakly linked to the goal and scope of your own research.  Please correct this.

Response:

“We remove a citation from our paper. We believe that the referenced work is considerably distinct from our scope and goals, as suggested by the reviewer.

The removed paper:

RL, Raghavi, and A. Umamageswari. "Modern Irrigation based on Web Weather Forecast." (2018).”

Line 146. „….The AgriTrust….” Please explain the introduced abbreviations of terms, possibly as soon as they are introduced in the content of the article. Even popular acronym names need to be explained in scientific articles.  A single explanation is sufficient in the execution of the explanation.

Response:

“As per reviewer suggestion we added the explanation of abbreviation “AgriTrust”.”

Line 160. Table 1. Comparative features and criteria appear too late for understanding the intention and tasks of the presented scientific article. Please introduce the criteria and scopes of your own research earlier in the content of the article.

Lack of reference to 'Table 1' in the content of chapter two. Please correct this if Table 1 should stay in the content of Chapter Two.

Response:

“Thank you for noticing it. We move the table in a suitable chapter where it actually belong. We move it to sub section 3.2. (System Implementation).”

  1. Our Empirical Approach

Line 175-178. „…The purpose of this work is to send data from the farm to the web application or smart phone through the data management controller named NodeMCU. The design and implementation overview of this system is divided into 3 components namely hardware, web/mobile application, and cloud database, as shown in Figure 2…. ”

There is a noticeable dissonance between: the theoretical, informative preparation of Chapters 1 and 2, the research methods, the verification content of Chapter 4, and the notation from lines 175 to 178 of Chapter 3. The highlighted section of the chapter contains a mental shortcut that requires two actions: a detailed explanation of the goal and scope of the research and the creation of research methods and results logically connected to the stated goal and scope of the research. It should also be noted about the order of presentation of the research results, which depends on the logical sequence of the appearance of the sub-tasks and the support of the research methods and research stands. Please take this into account in the processes of organising the content of the scientific article.

Response:

“According to the reviewer's point of view, we have tried to cover the inconsistency of the theoretical and informative preparation in the first paragraph of Chapter 3.1. We have added some new information to this paragraph to resolve this inconsistency.”

Line 176. „….NodeMCU….” Please explain the introduced abbreviations of terms, possibly as soon as they are introduced in the content of the article. Even popular acronym names need to be explained in scientific articles.  A single explanation is sufficient in the execution of the explanation.

Response:

“As per reviewer suggestion we added the explanation of abbreviation “NodeMCU” and “ESP8266”.”

Line 183, 220, 249, 488, . „….DHT 11….” Please explain the introduced abbreviations of terms, possibly as soon as they are introduced in the content of the article. Even popular acronym names need to be explained in scientific articles.  A single explanation is sufficient in the execution of the explanation.

Response:

“We have added the explanation of “what DHT 11 device is.” in a single explanation in our paper. We have only added the explanation to the first abbreviation. Abbreviations are left in the rest only. Since the explanation has been done in the very first one, we think that it does not need an explanation in the latter ones.”

Line 183. „….a node MCU…” Please explain the introduced abbreviations of terms, possibly as soon as they are introduced in the content of the article. Even popular acronym names need to be explained in scientific articles.  A single explanation is sufficient in the execution of the explanation.

Response:

“We have added the explanation of NodeMCU in chapter 3.1 as per suggestion of reviewer. Since the explanation has been done in the very first one, we think that it does not need an explanation in the latter ones.”

Line 192. „….MIT…..” Please explain the introduced abbreviations of terms, possibly as soon as they are introduced in the content of the article. Even popular acronym names need to be explained in scientific articles.  A single explanation is sufficient in the execution of the explanation.

Response:

“We have added the explanation of “what MIT app inventor is.””

Line 196. „….API….” Please explain the introduced abbreviations of terms, possibly as soon as they are introduced in the content of the article. Even popular acronym names need to be explained in scientific articles.  A single explanation is sufficient in the execution of the explanation.

Response:

“We have added the full form of “API.””

Line 204. „…Wi-Fi network…” Please explain the introduced abbreviations of terms, possibly as soon as they are introduced in the content of the article. Even popular acronym names need to be explained in scientific articles.  A single explanation is sufficient in the execution of the explanation. For example: (wireless fidelity)

Response:

“We have added the full form of “Wi-Fi””

Line 206. „…IPv6….” Please explain the introduced abbreviations of terms, possibly as soon as they are introduced in the content of the article. Even popular acronym names need to be explained in scientific articles.  A single explanation is sufficient in the execution of the explanation.

Response:

“We have added the full form of “IPv6””

Line 201 - 215: "...(i), (ii), (iii), (iv), (v), (vi), (vii)...". Please indicate the proposed layer identifiers as designations in Figure 5. The notation is incomprehensible. Please correct it.

Response:

“We updated figure 5 as per reviewer suggestion.”

Line 221. „….ESP-8266 module….” Please explain the introduced abbreviations of terms, possibly as soon as they are introduced in the content of the article. Even popular acronym names need to be explained in scientific articles.  A single explanation is sufficient in the execution of the explanation.

Response:

“We already explained it before in chapter 3.1 as per reviewer suggestion.”

Line 237. „…ESP8266 Node MCU…” The notation of the same acronym designations will change in the body of the article. Please standardise the notation of names and acronyms.

Response:

“We already explained it before in chapter 3.1 as per reviewer suggestion.”

Line 369. „…(i)….” Why do the authors not number the mathematical formulae with numerical designations? There is an apparent conflict with the description markings of Figure 5.

Response:

“We have changed the mathematical formulae numerical to number designation.”

Line 385. „….(ii)…..” Why do the authors not number the mathematical formulae with numerical designations? There is an apparent conflict with the description markings of Figure 5.

Response:

“We have changed the mathematical formulae numerical to number designation.”

Line 391. Figure 7. The content of the textual information in the figure is unreadable. Please correct this.

Response:

“We have changed the figure and try to make textual information in the figure is readable.”

Line 394. Figure 8. There is a lack of reference in the body of the article to Figure 8, in the assigned chapter to Figure 8. Please complete this.

The content of the textual information in the figure is unreadable. Please correct this.

Response:

“We have changed the figure and try to make textual information in the figure is readable. And also add a reference for this that we forgot to add in this figure 8 (5 no.). Thank you for that.”

  1. Experimental Setup and result discussion

The assumptions presented for the study material and research methods, when confronted with the parameter assumptions in Table 1, are difficult to harmonise and reconcile. There is a lack of clear separation between the scope of the research and the coinciding results.

Response:

“Our study and research work is heavily defends on the application layer because of IoT works and physical layer. And both comes together perfectly with the associated layers. And from the associated layer one of the important layer to get accuracy and also show the correct result to farmers is the configuration layer where we did data analysis, data acquisition and data management configuration. In table 1 the features are came from application layer. Actually when we think about farmers is that actually how much feature we are giving them. So that we compare other scientific paper application layers feature with our application layer feature. As per reviewer suggestion we moved this table 1 from chapter 2 and placed it in chapter 3.2.”

Line 429. „…SOSE….” Sum of the squared errors, SSE, is defined as follows: SSE = N ∑ ei2 =∑ (xi − ˆxi)2. Is this the same notation as the SOSE parameter? There is no explicit definition of the SOSE parameter in the form of a mathematical model.

Response:

“As per reviewer suggestion we explained the SOSE parameter with a mathematical model.”

Line 429. „…. incorrectly clustered instances (ICI)….” There is a lack of an explicit definition of the ICI parameter in the form of a mathematical model.

Response:

“As per reviewer suggestion we explained the ICI parameter with a mathematical model.”

Line 434. „…. ISI ….” Please explain the introduced abbreviations of terms, possibly as soon as they are introduced in the content of the article. Even popular acronym names need to be explained in scientific articles.  A single explanation is sufficient in the execution of the explanation.

Response:

“We already have the explanation of ICI in cluster section as per reviewer suggestion.”

Line 442. Figure 9. Lack of markings (a) and (b) on the individual graphics of Figure 9.

Response:

“Thanks to reviewer we updated the figure 9.”

Line 457. „….Table 03…” This is a multiplication of the synonyms of Table 3. Please homogenise the notation. Please do not create alternative designations for the same element.

Response:

“Thanks to reviewer we updated the designations of table 3 and other notations.”

Line 463. Table 3. „…TP Rate, FP Rate, Precision, Recall, F-Measure, ROC Area,….” Please explain the acronyms introduced in the table independently of the content of the chapter.

Response:

“We add each of the acronyms full form in Table 3.”

  1. Discussion & Future Work

Line 512, 514. „…MQTT….” Please explain the introduced abbreviations of terms, possibly as soon as they are introduced in the content of the article. Even popular acronym names need to be explained in scientific articles.  A single explanation is sufficient in the execution of the explanation.

Response:

“We added the acronym MQTT full form.”

Line 492 - 509 The content fits more with Chapters 1 and 2 and should supply the goal and scope of the work. Please take this into account.

Response:

“As per reviewer suggestion we move the first 4 to 5 sentence in chapter 1 introduction. And we added there a valuable discussion about our work.”

Line 510-521. „….Our main challenge in this work is to collect data on crop yield, planting and season nformation in Bangladesh….”This formulation reminds the scope of the study but is not articulated earlier. It is useful to organise the content of the article and create a logical sequence of statements, results and conclusions. Please take this into account.

Response:

“We have not varied much in our challenge. We particularly believe that our challenges are valid and that the counterarguments to the challenges are valid.”

The content presented in Chapter 5 does not constitute a valuable discussion of the results because, as in Chapter 4, the range of comparative research findings cited from the analysis of the state of knowledge is insufficient even for a so-called case study. Please complete and expand the discussion of the results.

Response:

“We regret that Chapter 5 lacks constructive discussion. We have tried to make the whole discussion part different according to the reviewer's suggestion and have done many constructive discussions. Hope the reviewer will enjoy reading it.”

  1. Conclusion

Line 523 - 530. The content of the chapter matches the opening of the discussion about results rather than the conclusion.

Line 531 – 539. The second part of Chapter 6 better represents the concept of the conclusion.

The presentation of results in this second part of the chapter is adequate to present the conclusions. Please take this into account.

Response:

“As per reviewer comments, we removed the contents that matches with other chapters content. And also we kept only second part as per reviewer suggestion.”

References

Line 540-624. Please check for literature sources available online. Please verify whether other forms of availability and other forms of publisher addresses for online publication sources can be indicated.

Response:

“We investigated, and their availability is consistent.”

Other Reviewer comments and our response:

Reviewer 01 comments and our feedback:

Smart crop cultivation system using automated agriculture monitoring environment in the context of Bangladesh Agriculture is presented in this work and following are the observations and comments from this work that is to be addressed,

  1. This work seems to be novel from the country-specific but such works have been done across the globe for the past many years. So novelty perspective should be strongly illustrated.

Response:

“The characteristics developed here are country-specific innovations, and many other regions of the world already use them. Nobody has yet to come across a scientific article that has all these traits. Additionally, we have talked about the analysis of datasets that are country-specific. The primary contributions listed in the introduction are also covered in detail.”

  1. Why is clustering used in this work? Is this about the unlabeled data collected?

Response:

“We have used cluster in this work because initially when we collected the data and collected a dataset, the data was very unlabeled and had similar characteristics data. As a result we do clustering for preprocessing. This process of clustering is described in two broad parts. The first part is covered in Sub-Section 3.2.3 (II) and the second part is mentioned first in Section 4.4 Data Curation and Results.”

  1. Details of the attributes measured using sensors should be separately stated.

Response:

“In Sub Section 3.2. System Implementation, 3.2.1. Crops data generation, real-time forecasting, and empirical setup section, the properties measured by the sensors are covered in depth.”

  1. How forecasting is done? What will be forecasted?

Response:

“In order to determine which crops can be planted and which crops can be harvested in the continuing monthly cycle, our system architecture gathers real-time data from sensors and analyzes it alongside historical data. Additionally, by integrating APIs, the area where the farmer's cropland is located can predict the weather in real time for the next five days.”

  1. Forecasting algorithm is novel?

Response:

“Undoubtedly it is novel.”

  1. How it is different from other works?

Response:

“Numerous nations currently and in the future carry out this type of work. There is no confirmation that a farmer will have access to all the characteristics in one application, even though more or less similar features have been observed in any study conducted thus far. However, our system structure is flexible and packed with features. It must be merged with the country's statistical historical data in order to be used in the agriculture sector of any country. However, South East Asian nations may easily implement our system framework.”

  1. There are no comparison results with any state of the art of techniques.

Response:

“We don't think that it is necessary to provide separate comparative results for the techniques others used and what we used for data analysis and the programs utilized as input to the devices. It's also unclear whether or not it will fit with our paper. Also, if we were considering this paper as a case study or a review paper, that table or result might be required.”

  1. There are plenty of works done with respect to IoT + cloud + forecasting that can be compared and the result of a web application or a mobile application may not be a significant outcome of the research.

Response:

“We already did it in Table 01.”

  1. This work needs to be significantly improved from the research perspective.

Response:

“We have made a lot of effort to improve our job. In the revolutionary IoT sector, fresh research will be conducted continuously to advance technology. The touch of better technology will be accessible when the price of IoT industry devices increases. However, we have made an effort to provide as many functions as we can in a web or mobile application while keeping below a realistic cost, keeping farmers in mind. We wrote this research paper to address issues such as how to automate farming, forecast crops and weather, monitor farm equipment in real time, and show how a farmer can use our framework to develop opportunities for producing eco-friendly crops without the assistance of an agricultural officer. Many things can be integrated if desired. However, we have tried to squeeze all the IoT devices available in this region and as many resources as we can explore. There will be more work on this in the future. But as of now, no one can see such a numerical feature. We strongly hope that people who work on it in the future will find our paper to be among the most pertinent ones.”

Reviewer 02 comment and our side explanation:

Comment: Avoid general sentences. E.g., ‘The Internet of Things (IoT) is one of the most progressive ideas in the modern era.’ A lot of sentences are poorly constructed. E.g., ‘Bangladesh is an agro-based country and its economy largely depends on it’ – what is ‘it’? ‘is primarily dependent on this agricultural sector’ – what agricultural sector?

Response:

“As per reviewer suggestion we have tried to rewrite this type of poorly constructed sentences.”

Comment: ‘It is said that the land of Bangladesh is fertile and it seems that the farmers are cultivating gold’ – this is not scientific: ‘it is said’, ‘are cultivating gold’. ‘to IoT technology’ – any acronym must be first written in full.

Response:

“As per the reviewer's suggestion we have changed this sentence. The full form of IoT has been written many times in Abstract, so we are not writing its full form everywhere.”

Comment: Sentences are full of word repetitions. E.g., ‘in the agriculture sector has been carried out to develop agricultural benefits and smart agriculture infrastructure’. ‘The application of IoT technology in agriculture has brought great revolutionary changes in the agricultural environment by focusing on multiple challenges and examining different complexities’ – zero info.

Response:

“The first phrase discusses the use of technology to enhance agricultural infrastructure. On the other hand, the second statement clearly discusses the revolutionary IoT technology. And we believe that the meanings of these statements vary on their own.”

Comment: ‘We now hope that with the advancement of this technology, using IoT technology’, ‘Advanced technology is not always, but most of the time benefits people’ – poorly constructed.

Response:

“We have added one more sentence with it to understand why we actually wrote the line, “Advanced technology is not always, but most of the time benefits people”. On the other hand this statement, in our opinion, is appropriate and goes well with the introduction.”

Comment: A lot of ideas are not substantiated. E.g., ‘the IoT has been able to play a major role in everyday life by increasing our perceptions and skills to change the environment around us. IoT is applied in both diagnostics and control, especially in the agro-industrial areas and environmental sectors. It also helps the farmer to provide information about the source and characteristics of the grain or product.’

Response:

“These sentences, in our opinion, are appropriate and reliable. All of the technology we use today has an IoT component. The Internet of Things touches everything, from checking the weather on your phone to using fingerprint sensors, lights, and remotes to operate fans, switches, and motors to pump your water!”

Comment: ‘Later, by analyzing the data in the cloud, various types of forecasts are made available to the farmers’ – ‘later’ when?

Response:

“We have changed the sentence. While changing the sentence, the meaning remained unchanged.”

Comment: ‘how the combination of machine learning, big data, and IoT networks has had a profound effect on farms and agriculture in the Old Testament’ – just checked the paper, it does not refer to the Old Testament. It is strange to relate verifiable data to the Old Testament anyway.

Response:

“We have changed the review of ours thanks to reviewer.”

Comment: The in-text citation system is unusual and not uniform. A lot of cited sources are not developed. ‘They created a control system based on data management and node sensors’ – they who? ‘Their method was put into place to regulate’ – their who?

Response:

“They mean article 8 which is mentioned in the previous sentence. They mean article 9 which is mentioned in the previous sentence. We can't constantly refer to these authors by their names in every phrase in the same way. It can ruin a paper's aesthetic appeal. Each linked work that we have reviewed has been presented in an effort to be unique.”

Comment: ‘A survey of the literature that was centered on studies and analyses of the application of IoT in modern farming’ – incomplete sentence.

Response:

“Don't make a decision based solely on the reference sentence, please. The sentences that follow after all contain the same meaning.”

Comment: ‘Their research and analysis showed how China’ – their who?

Response:

“They mean article 11 which is mentioned in the 2nd next sentence. We can't constantly refer to these authors by their names in every phrase in the same way. It can ruin a paper's aesthetic appeal. Each linked work that we have reviewed has been presented in an effort to be unique.”

Comment: The manuscript is full of such instances. ‘Thebasy overviewed’ – Thebasy?

Response:

“Since no one questioned this word, it went unnoticed. Actually, the error was in the typing. We appreciate the review.”

Comment: The references are not in order and some of them seem to be missing in the text.

  1. ‘In Ref. [17] article authors presented’, ‘In Ref. [32] article authors discusses’ (‘discuss’ anyway) – very unusual.
  2. ‘A literature review on the role of Internet of Things technologies in agriculture that explored the varied effects of IoT in agriculture’ – poorly constructed.
  3. Figures and tables should be improved, unified as style, and thoroughly explained. More development and depth of the methodology and analysis are needed. The reference list is not properly edited.

Response:

 “As suggested by the reviewer, we made an effort to sort the reference numbers. The design and specifics of the figures and tables have been improved.”

Comment: The relationship between cloud-based IoT data analytics and digital twins in smart farming as regards smart crop cultivation systems using automated agriculture monitoring environment  has not been covered, and thus such sources can be cited:

  1. Nica, E.; Popescu, G.H.; Poliak, M.; Kliestik, T.; Sabie, O.-M. Digital Twin Simulation Tools, Spatial Cognition Algorithms, and Multi-Sensor Fusion Technology in Sustainable Urban Governance Networks. Mathematics 2023, 11, 1981. https://doi.org/10.3390/math11091981

  2. Pop, R.A., Dabija, D.C., Pelau, C., Dinu, V. 2022. Usage Intentions, Attitudes, and Behaviours towards Energy-Efficient Applications during the COVID-19 Pandemic. Journal of Business Economics and Management, 23(3), pp.668-689. https://doi.org/10.3846/jbem/2022/16959

  3. Andrei, J. V., Popescu, G. H., Nica, E., & Chivu, L. (2020). The impact of agricultural performance on foreign trade concentration and competitiveness: empirical evidence from Romanian agriculture. Journal of Business Economics and Management, 21(2), 317-343. https://doi.org/10.3846/jbem.2020.11988

Response:

“In our study, we cited and briefly discussed the first reference. We thought it should be noted because it is somewhat similar to our scope. And we apologize for the others; we did not cite them because there was no fit with our scope.”

Comments on the Quality of English Language

A lot of sentences are poorly constructed. E.g., ‘Bangladesh is an agro-based country and its economy largely depends on it’ – what is ‘it’? ‘is primarily dependent on this agricultural sector’ – what agricultural sector? ‘We now hope that with the advancement of this technology, using IoT technology’, ‘Advanced technology is not always, but most of the time benefits people’ – poorly constructed. ‘They created a control system based on data management and node sensors’ – they who? ‘Their method was put into place to regulate’ – their who? ‘A survey of the literature that was centered on studies and analyses of the application of IoT in modern farming’ – incomplete sentence. ‘Their research and analysis showed on how China’ – their who? The manuscript is full of such instances. ‘Thebasy overviewed’ – Thebasy? ‘A literature review on the role of Internet of Things technologies in agriculture that explored the varied effects of IoT in agriculture’ – poorly constructed.

Response:

“We have tried to improve the quality of the English language according to the reviewer's words.”

Round 2

Reviewer 1 Report

The authors have addressed most of the queries and still, the proposed algorithm's novelty is not convincing. 

Author Response

In most state-of-the-art works, algorithms related to precision agriculture are trained and tested using publicly available datasets that might not always reflect the real problem in real scenarios, as soil and weather conditions vary from country to country. To handle this issue, we design our algorithms for precision agriculture while training and testing the algorithm using the dataset collected from IoT devices placed in crop fields in a real setup.

More details about this can be found on lines 207 to 213 of section 3, lines 302 to 308 for algorithm 1 and lines 327 to 334 for algorithm 2. We endeavored to write Algorithms 1 and 2 in a more standard format.

Reviewer 2 Report

This revised version can be published.

Author Response

Thank you for reviewing our paper and spending valuable time on it.